# Role of the transcriptional regulator SP140 in resistance to bacterial infections via repression of type I interferons

Daisy X Ji[1†], Kristen C Witt[1†], Dmitri I Kotov[1,2], Shally R Margolis[1], Alexander Louie[1], Victoria Chevée[1], Katherine J Chen[1,2], Moritz M Gaidt[1], Harmandeep S Dhaliwal[3], Angus Y Lee[3], Stephen L Nishimura[4], Dario S Zamboni[5], Igor Kramnik[6], Daniel A Portnoy[1,7,8], K Heran Darwin[9], Russell E Vance[1,2,3]*

[1]Division of Immunology and Pathogenesis, Department of Molecular and Cell Biology, University of California, Berkeley, Berkeley, United States; [2]Howard Hughes Medical Institute, University of California, Berkeley, Berkeley, United States; [3]Cancer Research Laboratory, University of California, Berkeley, Berkeley, United States; [4]Department of Pathology, University of California, San Francisco, San Francisco, United States; [5]Department of Cell Biology, Ribeirão Preto Medical School, University of São Paulo, São Paulo, Brazil; [6]The National Emerging Infectious Diseases Laboratory, Department of Medicine (Pulmonary Center), and Department of Microbiology, Boston University School of Medicine, Boston, United States; [7]Division of Biochemistry, Biophysics and Structural Biology, Department of Molecular and Cell Biology, University of California, Berkeley, Berkeley, United States; [8]Department of Plant and Microbial Biology, University of California, Berkeley, Berkeley, United States; [9]Department of Microbiology, New York University Grossman School of Medicine, New York, United States

*For correspondence: rvance@berkeley.edu

[†]These authors contributed equally to this work

**Abstract** Type I interferons (IFNs) are essential for anti-viral immunity, but often impair protective immune responses during bacterial infections. An important question is how type I IFNs are strongly induced during viral infections, and yet are appropriately restrained during bacterial infections. The *Super susceptibility to tuberculosis 1* (*Sst1*) locus in mice confers resistance to diverse bacterial infections. Here we provide evidence that *Sp140* is a gene encoded within the *Sst1* locus that represses type I IFN transcription during bacterial infections. We generated *Sp140*[−/−] mice and found that they are susceptible to infection by *Legionella pneumophila* and *Mycobacterium tuberculosis*. Susceptibility of *Sp140*[−/−] mice to bacterial infection was rescued by crosses to mice lacking the type I IFN receptor (*Ifnar*[−/−]). Our results implicate *Sp140* as an important negative regulator of type I IFNs that is essential for resistance to bacterial infections.

## Introduction

Type I interferons (IFNs) comprise a group of cytokines, including interferon-β and multiple interferon-α isoforms, that are essential for immune defense against most viruses (*Stetson and Medzhitov, 2006*). Type I IFNs signal through a cell surface receptor, the interferon alpha and beta receptor (IFNAR), to induce an 'anti-viral state' that is characterized by the transcriptional induction of hundreds of interferon-stimulated genes (ISGs) (*Schneider et al., 2014*). Many ISGs encode proteins with direct anti-viral activities. Type I IFNs also promote anti-viral responses by cytotoxic T cells and natural killer cells. Accordingly, *Ifnar*[−/−] mice are highly susceptible to most viral infections.

Many ISGs are also induced by IFN-γ (also called type II IFN). However, type I and type II IFNs appear to be specialized for the control of different classes of pathogens (*Crisler and Lenz, 2018*). Whereas type I IFNs are predominantly anti-viral, the ISGs induced by IFN-γ appear to be especially important for the control of diverse intracellular pathogens, including parasites and bacteria. In contrast, type I IFNs play complex roles during bacterial infections (*Boxx and Cheng, 2016*; *Donovan et al., 2017*; *McNab et al., 2015*; *Moreira-Teixeira et al., 2018*). Some ISGs induced by type I IFN, most notably certain guanylate-binding proteins (GBPs), have anti-bacterial activities (*Pilla-Moffett et al., 2016*). At the same time, other proteins induced by type I IFNs, including interleukin-10 (IL-10) and IL-1 receptor antagonist (IL-1RA), impair anti-bacterial immunity (*Boxx and Cheng, 2016*; *Ji et al., 2019*; *Mayer-Barber et al., 2014*). As a result, the net effect of type I IFN is often to increase susceptibility to bacterial infections. For example, $Ifnar^{-/-}$ mice exhibit enhanced resistance to *Listeria monocytogenes* (*Auerbuch et al., 2004*; *Carrero et al., 2004*; *O'Connell et al., 2004*) and *Mycobacterium tuberculosis* (*Donovan et al., 2017*; *Dorhoi et al., 2014*; *Ji et al., 2019*; *Mayer-Barber et al., 2014*; *Moreira-Teixeira et al., 2018*). Multiple mechanisms appear to explain resistance of $Ifnar^{-/-}$ mice to *L. monocytogenes*, including a negative effect of type I IFNs on protective IFN-γ signaling (*Rayamajhi et al., 2010*). Likewise, diverse mechanisms underlie the negative effects of type I IFNs during *M. tuberculosis* infection, including alterations of eicosanoid production (*Mayer-Barber et al., 2014*) and the induction of IL-1Ra (*Ji et al., 2019*), both of which impair protective IL-1 responses.

As an experimental model for dissecting the mechanisms by which inappropriate type I IFN responses are restrained during bacterial infections, we have compared mice harboring different haplotypes of the *Super susceptibility to tuberculosis 1* (*Sst1*) locus (*Pan et al., 2005*; *Pichugin et al., 2009*). The *Sst1* locus encompasses about 10M base pairs of mouse chromosome 1, a region that contains approximately 50 genes. Mice harboring the susceptible (S) haplotype of *Sst1*, derived from the C3H/HeBFeJ mouse strain, succumb relatively rapidly to *M. tuberculosis* infection as compared to isogenic mice harboring the resistant (R) *Sst1* haplotype (derived from C57BL/6 mice). Likewise, $Sst1^S$ mice also exhibit enhanced susceptibility to *L. monocytogenes* (*Boyartchuk et al., 2004*; *Pan et al., 2005*) and *Chlamydia pneumoniae* (*He et al., 2013*). The susceptibility of $Sst1^S$ mice to *M. tuberculosis* was reversed by crossing to $Ifnar^{-/-}$ mice (*Ji et al., 2019*), thereby demonstrating the causative role of type I IFNs in driving the susceptibility phenotype. Although multiple type I IFN-induced genes are likely responsible for the detrimental effects of type I IFNs during bacterial infections, heterozygous deficiency of a single type I IFN-induced gene, *Il1rn* (encoding IL-1 receptor antagonist), was sufficient to almost entirely reverse the susceptibility of $Sst1^S$ mice to *M. tuberculosis* (*Ji et al., 2019*).

The $Sst1^R$ haplotype is dominant over the $Sst1^S$ haplotype, suggesting that $Sst1^R$ encodes a protective factor that is absent from $Sst1^S$ mice (*Pan et al., 2005*; *Pichugin et al., 2009*). By comparing gene expression in $Sst1^R$ versus $Sst1^S$ mice, *Sp110* (also known as *Ipr1*) was discovered as an *Sst1*-encoded gene that is transcribed selectively in $Sst1^R$ mice (*Pan et al., 2005*). Transgenic expression of *Sp110* in $Sst1^S$ mice partially restored resistance to *M. tuberculosis* and *L. monocytogenes* (*Pan et al., 2005*). However, the causative role of *Sp110* in conferring resistance to bacterial infections was not confirmed by the generation of *Sp110*-deficient B6 mice. Null mutations of human *SP110* are associated with VODI (hepatic veno-occlusive disease with immunodeficiency syndrome, OMIM 235550), but not mycobacterial diseases (*Roscioli et al., 2006*). Some studies have found polymorphisms in *SP110* to be associated with susceptibility to TB, though not consistently so across different ethnic groups (*Chang et al., 2018*; *Fox et al., 2014*; *Lei et al., 2012*; *Png et al., 2012*; *Thye et al., 2006*; *Tosh et al., 2006*; *Zhang et al., 2017*).

In humans and mice, SP110 is a part of the speckled protein (SP) family of nuclear proteins, consisting of SP100, SP110, and SP140 (and SP140L in humans only) (*Fraschilla and Jeffrey, 2020*). The SP family members also exhibit a high degree of similarity to AIRE, a transcriptional regulator that promotes tolerance to self-antigens by inducing their expression in thymic epithelial cells (*Anderson and Su, 2016*; *Fraschilla and Jeffrey, 2020*; *Perniola and Musco, 2014*). All members of the SP-AIRE family in both mice and humans have an N-terminal SP100 domain that appears to function as a homotypic protein-protein interaction domain (*Fraschilla and Jeffrey, 2020*; *Huoh et al., 2020*). The SP100 domain is closely related to the caspase activation and recruitment domain (CARD), though SP family members are not believed to activate caspases. SP-AIRE proteins also contain a DNA-binding SAND domain (*Bottomley et al., 2001*). Certain SP isoforms, including

all human full-length SP family members and mouse SP140, also include a plant homeobox domain (PHD) and a bromodomain (BRD) (*Fraschilla and Jeffrey, 2020*). The genes encoding SP family proteins are linked in a small cluster in both mouse and human genomes and are inducible by IFN-γ in a variety of cell lines. The mouse *Sp100/110/140* gene cluster is adjacent to a highly repetitive 'homogenously staining region' (HSR) of chromosome 1 that remains poorly assembled in the most recent genome assembly due to the presence of as many as 40 near-identical repeats of *Sp110*-like sequences (*Pan et al., 2005*; *Weichenhan et al., 2001*). Most of these repeated *Sp110*-like sequences in the HSR appear to be either incomplete copies of *Sp110* or pseudogenes that are not believed to be translated, but their presence has nevertheless complicated genetic targeting and analysis of the SP gene family.

With the advent of CRISPR/Cas9-based methods (*Wang et al., 2013*), we were able to generate $Sp110^{-/-}$ mice on the B6 background. Surprisingly, we found that $Sp110^{-/-}$ mice do not phenocopy the susceptibility of $Sst1^S$ mice to *M. tuberculosis* infection in vivo. Upon analysis of additional candidate genes in the *Sst1* locus, we found that B6.$Sst1^S$ mice also lack expression of *Sp140*. To test whether loss of *Sp140* might account for the susceptibility of $Sst1^S$ mice to bacterial infections, we generated $Sp140^{-/-}$ mice. We found that these mice are as susceptible as B6.$Sst1^S$ mice to the intracellular bacterial pathogens *M. tuberculosis* and *Legionella pneumophila*. Similar to B6.$Sst1^S$ mice, $Sp140^{-/-}$ mice exhibit an exacerbated type I IFN response after bacterial infection, and the susceptibility of $Sp140^{-/-}$ mice is rescued by crosses to $Ifnar^{-/-}$ mice. Our results suggest that loss of *Sp140* explains the susceptibility to bacterial infections associated with the $Sst1^S$ haplotype. These data further suggest that SP140 is a novel negative regulator of type I IFN responses that is essential for protection against intracellular bacterial infections.

## Results

### $Sp110^{-/-}$ mice are not susceptible to *M. tuberculosis*

Loss of *Sp110* expression was proposed to account for the susceptibility of mice carrying the $Sst1^S$ haplotype to bacterial infections (*Pan et al., 2005*). We first confirmed that bone marrow-derived macrophages (BMMs) from B6.$Sst1^S$ mice lack expression of *Sp110* protein (*Figure 1A*). To determine whether loss of *Sp110* confers susceptibility to bacterial infections, we used CRISPR/Cas9 to target exon 3 of *Sp110* to generate $Sp110^{-/-}$ mice on the C57BL/6 (B6) background (*Figure 1—figure supplement 1*). We generated three independent $Sp110^{-/-}$ lines, denoted as lines 61, 65, and 71 (*Figure 1A*, *Figure 1—figure supplement 1*). All three lines lacked expression of *Sp110*, as verified using three different antibodies (*Figure 1A*). $Sp110^{-/-}$ mice are viable and are born at normal Mendelian ratios and litter sizes. When aerosol infected with a low dose of *M. tuberculosis*, $Sp110^{-/-}$ mice did not phenocopy the susceptibility observed in B6.$Sst1^S$ mice (*Figure 1B-D*). At day 25 post-infection, $Sp110^{-/-}$ lungs resembled those of wild-type B6 mice (*Figure 1B*) and harbored fewer bacteria than the lungs of B6.$Sst1^S$ mice, similar to both the B6 and $Sp110^{+/-}$ littermates (*Figure 1C*). Likewise, the survival of infected $Sp110^{-/-}$ mice was indistinguishable from B6 mice, and mice of both genotypes survived considerably longer than the B6.$Sst1^S$ mice (*Figure 1D*). Thus, despite the absence of *Sp110* from $Sst1^S$ mice, our results indicate that the loss of *Sp110* is not sufficient to replicate the susceptibility to *M. tuberculosis* associated with the $Sst1^S$ locus.

### $Sp140^{-/-}$ mice are susceptible to bacterial infections

Given that *Sp110* deficiency did not phenocopy the susceptibility of $Sst1^S$ mice, we asked whether any other genes found within the *Sst1* locus differ in expression between B6 and B6.$Sst1^S$ BMMs. We noted that a homolog of *Sp110* called *Sp140* was also reduced in expression in B6.$Sst1^S$ cells compared to B6 cells (*Figure 2A*). Immunoblot confirmed that IFN-γ treated BMMs from B6.$Sst1^S$ mice do not produce SP140 protein (*Figure 2B*). We therefore used CRISPR/Cas9 to generate two independent lines of $Sp140^{-/-}$ mice on a pure B6 background (*Figure 2—figure supplement 1A-C*). Our analysis focused primarily on line 1, which we found lacked expression of SP140 protein (*Figure 2B*) but retains the production of SP110 protein (*Figure 2—figure supplement 1D*). Like $Sp110^{-/-}$ and $Sst1^S$ mice, $Sp140^{-/-}$ mice are viable, fertile, and born at the expected Mendelian ratios. When infected with *M. tuberculosis*, however, $Sp140^{-/-}$ mice exhibited high bacterial burdens

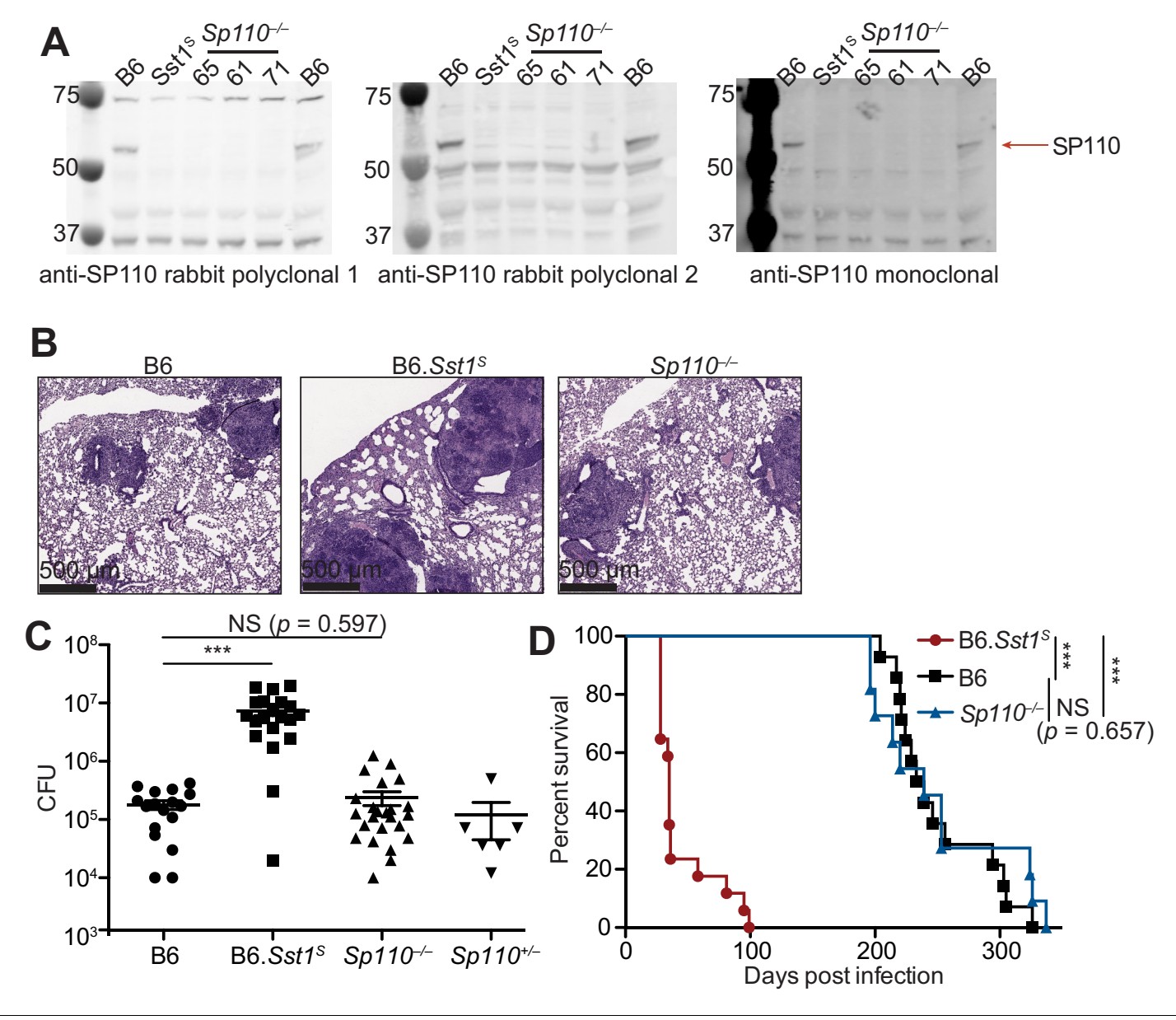

Figure 1. *Sp110*⁻/⁻ mice are not susceptible to *Mycobacterium tuberculosis* infections. (A) BMMs were treated with 10 U/ml of IFNγ for 24 hr and cells were lysed with RIPA buffer. Five micrograms of total protein was loaded on each lane, and immunoblot was performed with respective antibodies as shown. Molecular weight standards are shown on the left of each blot in kDa. Individual membranes were imaged separately. Three independent lines of *Sp110*⁻/⁻ mice were analyzed (denoted lines 61, 65, and 71). (B–D) Lungs of mice infected with *M. tuberculosis* were stained with hematoxylin and eosin (H&E) for histology (B), measured for CFU at 25 days post-infection (Mann-Whitney test) (C), or monitored for survival (D). All except B6 mice were bred in-house, and combined results from the three independent *Sp110*⁻/⁻ lines are shown. Representative of two experiments (B, D); combined results of three infections (C). *p≤0.05; **p≤0.01; ***p≤0.005. BMM, bone marrow-derived macrophage; CFU, colony-forming unit; RIPA, radioimmunoprecipitation assay.

The online version of this article includes the following figure supplement(s) for figure 1:

**Figure supplement 1.** CRISPR/Cas9 targeting strategy for *Sp110*⁻/⁻ mice.

in their lungs, similar to B6.*Sst1*$^S$ mice and significantly greater than B6, *Sp110*⁻/⁻ or *Sp140*⁺/⁻ litter-mate mice at day 28 post-infection (*Figure 2C*, *Figure 2—figure supplement 1D*).

We performed hematoxylin and eosin (H&E) staining of lung sections from B6, *Sp140*⁻/⁻, *Sp110*⁻/⁻, and B6.*Sst1*$^S$ mice 25 days after *M. tuberculosis* infection, and qualitatively assessed

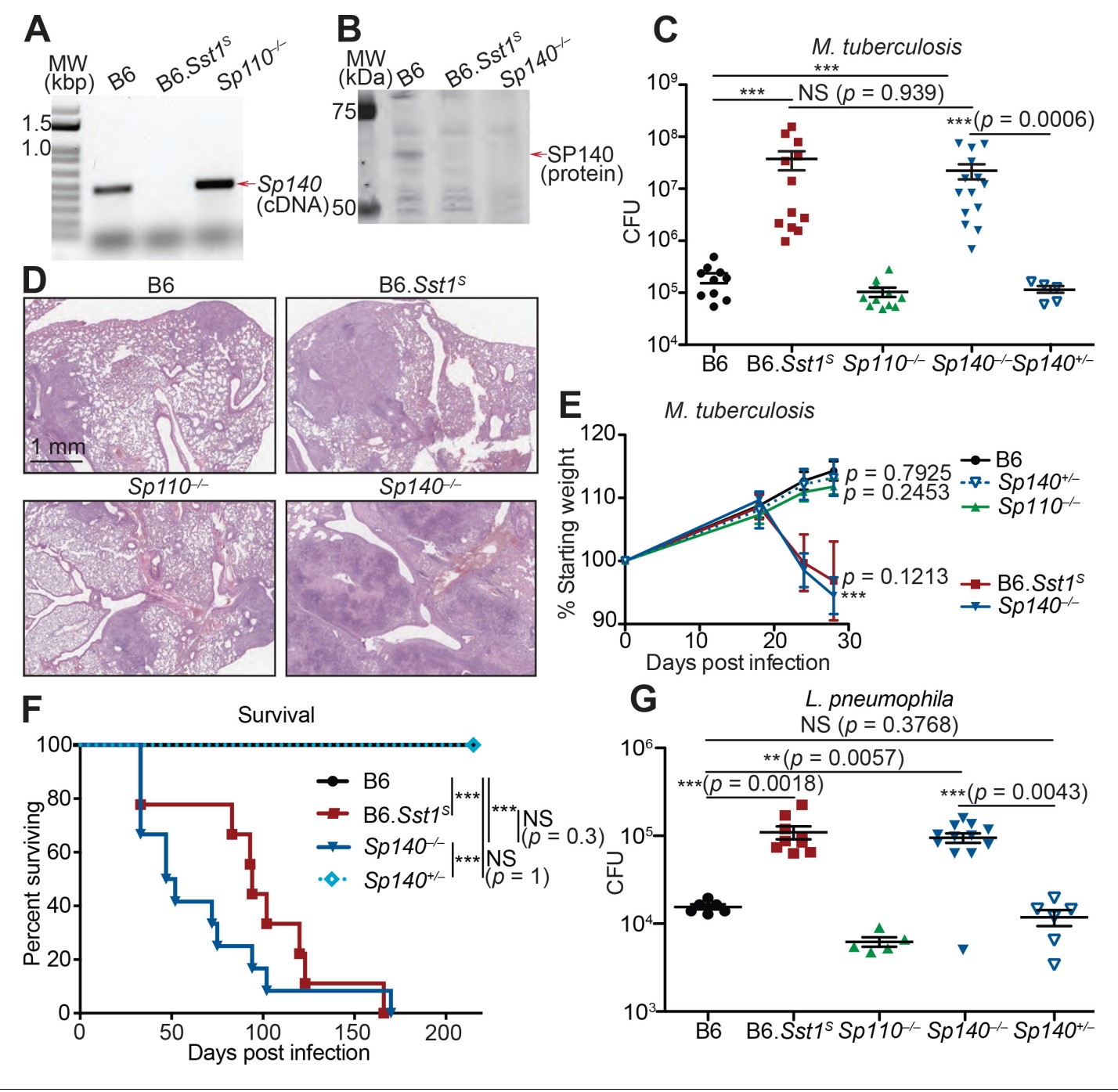

**Figure 2.** $Sp140^{-/-}$ mice are susceptible to bacterial pathogens. (A) RT-PCR of cDNA from BMMs of the indicated genotypes. Red arrow indicates band corresponding to a portion of $Sp140$, verified by sequencing. (B) Immunoblot of lysates from $Sp140^{-/-}$ and B6 BMMs treated with 10 U/ml of recombinant mouse IFNγ for 24 hr. Equal amounts of protein were loaded for immunoblot with anti-SP140 antibody. (C–F) Mice were infected with *Mycobacterium tuberculosis* and measured for (C) lung CFU at 28 days post-infection, (E) body weight over time, and (F) survival. Statistics in (E) shows the comparison to B6 at day 28, and data are from 10 B6, 11 $B6.Sst1^S$, 11 $Sp110^{-/-}$, 14 $Sp140^{-/-}$, and 6 $Sp140^{+/-}$ mice. (D) H&E staining of lungs at 25 days post-infection with *M. tuberculosis*. Full histology images are provided in *Figure 2—figure supplement 2*. (G) Mice were infected with *Legionella pneumophila* and lung CFUs were determined at 96 hr post-infection. All mice were bred in-house, $Sp140^{-/-}$ and $Sp140^{+/-}$ were littermates (C–F). (C), (E), and (G) are combined results of two independent infections. (A–D) show representative analysis of one $Sp140^{-/-}$ line (line 1), whereas (F, G) include a mixture of both lines 1 and 2. Results of infection of both lines with *M. tuberculosis* are shown in *Figure 2—figure supplement 1E*. (C, E, F, G) Mann-Whitney test. *p≤0.05; **p≤0.01; ***p≤0.005. BMM, bone marrow-derived macrophage; CFU, colony-forming unit; H&E, hematoxylin and eosin; RT-qPCR, real-time quantitative-polymerase chain reaction; WT, wild-type.

*Figure 2 continued on next page*

*Figure 2 continued*

The online version of this article includes the following figure supplement(s) for figure 2:

**Figure supplement 1.** CRISPR/Cas9 targeting strategy for *Sp140*$^{-/-}$ and validation of founders.

**Figure supplement 2.** Histology of lungs from B6, B6.*Sst1*$^S$, *Sp110*$^{-/-}$, and *Sp140*$^{-/-}$ mice after infection with *Mycobacterium tuberculosis*.

**Figure supplement 3.** Characterization of off-target genes mutated in *Sp140*$^{-/-}$ mice.

**Figure supplement 4.** Complementation of hyper type I IFN responses in *Sp140*$^{-/-}$ BMMs.

macrophage, lymphoid, and granulocyte infiltration as well as the extent of necrosis (*Figure 2D*, *Figure 2—figure supplement 2*). We found that *Sp140*$^{-/-}$ and B6.*Sst1*$^S$ lungs showed moderately increased granulocyte infiltration by 25 days post infection, with apparently more severe infiltration in *Sp140*$^{-/-}$ mice than in B6.*Sst1*$^S$ mice, though this difference is not statistically significant. We could also discern some areas of necrosis in the *Sp140*$^{-/-}$ lungs, although our samples were taken at an early timepoint that precedes the formation of hypoxic lesions observed in B6.*Sst1*$^S$ lungs upon *M. tuberculosis* infection (*Harper et al., 2012*). The increased susceptibility of *Sp140*$^{-/-}$ mice was accompanied by significant weight loss and shortened survival upon infection with *M. tuberculosis*, again phenocopying the B6.*Sst1*$^S$ mice (*Figure 2E-F*). Both of the independent lines of *Sp140*$^{-/-}$ mice were similarly susceptible to *M. tuberculosis* (*Figure 2—figure supplement 1E*). We also found that both B6.*Sst1*$^S$ and *Sp140*$^{-/-}$ mice were more susceptible to the intracellular Gram-negative bacterium *L. pneumophila*, as compared to the B6 and *Sp110*$^{-/-}$ mice (*Figure 2G*).

An important caveat to the use of CRISPR/Cas9 to generate *Sp140*$^{-/-}$ mice is the presence of an unknown number of nearly identical *Sp140*-like genes in the *Sst1* locus and non-localized chromosome 1 genome contigs (that presumably map to the adjacent HSR that remains unassembled by the mouse genome project). It is possible that the guide RNA we used to disrupt exon 3 of *Sp140* also disrupted these uncharacterized *Sp140*-like genes, though it is not clear if these uncharacterized *Sp140*-like genes give rise to functional proteins. Nevertheless, to identify potential mutated off-target genes in our *Sp140*$^{-/-}$ mice, we amplified exons 2/3 of *Sp140* and any potential paralogs from genomic DNA and from cDNA derived from *M. tuberculosis*-infected lungs, and subjected the amplicons to deep sequencing (*Figure 2—figure supplement 3*). Although we found evidence for several edited *Sp140*-like exons in our *Sp140*$^{-/-}$ mice, only one of these edited off-target genes was found to be detectably expressed from analysis of RNA-seq data from *M. tuberculosis*-infected lungs, and this off-target appeared to be edited in only one of our founder lines (*Figure 2—figure supplement 3B*). Thus, mutation of *Sp140* itself is the most parsimonious explanation for susceptibility of our *Sp140*$^{-/-}$ mice, a conclusion further supported by complementation of the mutation in BMMs (see below, *Figure 2—figure supplement 4*). Collectively our results strongly suggest that the lack of expression of *Sp140* in B6.*Sst1*$^S$ mice explains the broad susceptibility of these mice to bacterial infections.

## Enhanced type I IFN responses in *Sp140*$^{-/-}$ and B6.*Sst1*$^S$ mice

We and others previously reported that TNF$\alpha$ induces higher levels of type I IFN-induced genes in *Sst1*$^S$ BMMs as compared to B6 BMMs (*Bhattacharya et al., 2021*; *Ji et al., 2019*). We also observed higher levels of *Ifnb* transcripts in the lungs of B6.*Sst1*$^S$ mice infected with *M. tuberculosis*, as compared to infected B6 mice (*Ji et al., 2019*). Similar to B6.*Sst1*$^S$ BMMs, *Sp140*$^{-/-}$ BMMs also exhibited elevated expression of *Ifnb* and ISGs when stimulated with TNF$\alpha$ (*Figure 2—figure supplement 4A*). Importantly, we were also able to complement the enhanced IFN phenotype of *Sp140*$^{-/-}$ BMMs by transducing *Sp140*$^{-/-}$ BMMs with a retrovirus expressing an *Sp140* cDNA driven by a minimal CMV promoter (*Figure 2—figure supplement 4B*). Repression of *Ifnb* by overexpression of *Sp140* in *Sp140*$^{-/-}$ BMMs was selective, as *Sp140* overexpression did not repress the transcription of *Tnfa* induced by TNF$\alpha$ (*Figure 2—figure supplement 4B*).

In addition to enhanced type I IFN responses to TNF$\alpha$, we also observed that both B6.*Sst1*$^S$ and *Sp140*$^{-/-}$ BMMs show increased cell death in vitro upon stimulation with polyI:C compared to B6 BMMs, as measured by lactate dehydrogenase (LDH) release (*Figure 3—figure supplement 1*). This result is analogous to previous findings that B6.*Sst1*$^S$ BMMs die upon sustained TNF stimulation (*Brownhill et al., 2020*). The enhanced polyI:C-induced LDH release in both *Sp140*$^{-/-}$ and B6.*Sst1*$^S$

BMMs was blunted upon genetic deletion of *Ifnar* (*Figure 3—figure supplement 1*), consistent with type I IFNs playing an important role in the cell death phenotype.

When infected with *M. tuberculosis*, the lungs of *Sp140*$^{-/-}$ and B6.*Sst1*$^S$ mice also exhibited higher levels of *Ifnb* transcript as compared to B6, *Sp110*$^{-/-}$, and *Sp140*$^{+/-}$ littermate mice (*Figure 3A*). The lungs of *Sp140*$^{-/-}$ exhibited moderately increased levels of *Ifnb* transcript compared to B6.*Sst1*$^S$ during *M. tuberculosis* infection, which could be a result of partial low expression of *Sp140* in B6.*Sst1*$^S$ mice, or possibly microbiota differences between the strains. Since we do not observe significant differences in weight, survival, or colony-forming unit (CFU) between *Sp140*$^{-/-}$ and B6.*Sst1*$^S$ mice upon *M. tuberculosis* infection, there is no evidence that modest differences in type I IFN responses are of functional significance. We also found that during *L. pneumophila* infection, *Sp140*$^{-/-}$ mice expressed more *Ifnb* in their lungs, as compared to B6 mice (*Figure 3B*). Importantly, elevated *Ifnb* was evident at 48 hr post-infection when there is no difference in bacterial burdens between the genotypes, and at 96 hr post-infection, when *Sp140*$^{-/-}$ mice have greater bacterial burdens (*Figure 3B*).

## Infected *Sp140*$^{-/-}$ and B6.*Sst1*$^S$ lungs show similar gene expression patterns

We used RNA sequencing to analyze the global gene expression patterns in *M. tuberculosis*-infected lungs of B6, *Sp110*$^{-/-}$, *Sp140*$^{-/-}$, and B6.*Sst1*$^S$ mice at day 28 post-infection (*Figure 4*). Principal component analysis revealed that while there is spread among individual samples, the expression pattern of *Sp140*$^{-/-}$ and B6.*Sst1*$^S$ lungs segregates from the expression pattern in B6 and *Sp110*$^{-/-}$ lungs along the PC1 axis (77% of variance) (*Figure 4A*). Notably, the *Sp140*$^{-/-}$ and B6.*Sst1*$^S$ only separated along the PC2 axis, which accounts for only 9% of the variance in our RNA-seq data. Euclidean distance analysis revealed a similar pattern, with B6.*Sst1*$^S$ and *Sp140*$^{-/-}$ mice clustering together, and away from B6 and *Sp110*$^{-/-}$ mice (*Figure 4B*). At the time point analyzed (28 dpi), both *Sp140*$^{-/-}$ and B6.*Sst1*$^S$ mice exhibit higher bacterial burdens than B6 and *Sp110*$^{-/-}$ mice (*Figure 2C*). Thus, the similarity of the gene expression profile of B6.*Sst1*$^S$ and *Sp140*$^{-/-}$ lungs may merely reflect increased inflammation in these lungs. Alternatively, the increased bacterial burdens may be due to a similarly enhanced type I IFN response in these mice, which leads to secondary bacterial outgrowth and inflammation. Therefore, we specifically compared the change in expression of two subsets of genes: (1) hallmark inflammatory response pathway (*Figure 4C*) and (2) type I interferon response genes (*Figure 4D*). This analysis revealed that B6.*Sst1*$^S$ and *Sp140*$^{-/-}$ mice not only show a similarly increased inflammatory gene signature, as expected, but in addition showed a similarly increased type I IFN gene signature. We validated the elevated expression of the ISG *Il1rn* in both B6.*Sst1*$^S$ and *Sp140*$^{-/-}$ mice during *M. tuberculosis* infection by real-time quantitative polymerase chain reaction (RT-qPCR) (*Figure 4—figure supplement 1*). The expression of *Sp110* and the SP family member *Sp100* in *Sp140*$^{-/-}$ mice during *M. tuberculosis* infection was unimpaired compared to B6, and the expression level of *Sp100* was unchanged between *Sp140*$^{-/-}$ and B6.*Sst1*$^S$ mice (*Figure 4—figure supplement 2*). We also did not observe major changes in expression (>2-fold change) of *Sp100* or *Sp140* in *Sp110*$^{-/-}$ mice during *M. tuberculosis* infection (*Figure 4—figure supplement 2*). Overall, the expression of additional SP family members in *Sp140*$^{-/-}$ and *Sp110*$^{-/-}$ mice is intact, which suggests that the targeting of these genes had specific rather than unanticipated epistatic effects. Therefore, deficiency in *Sp140* is likely the primary driver of susceptibility in *Sp140*$^{-/-}$ mice, while the resistance of *Sp110*$^{-/-}$ mice likely derives from normal expression of *Sp140* rather than aberrant changes in the expression of other SP family members.

Only 269 genes were significantly differentially expressed (adjusted p-value <0.05) between *Sp140*$^{-/-}$ and B6.*Sst1*$^S$ samples, whereas 1520 genes were significantly differentially expressed between *Sp140*$^{-/-}$ and B6. Within the 269 genes differentially expressed between *Sp140*$^{-/-}$ and B6.*Sst1*$^S$, 62 were immunoglobulin genes, 62 were annotated as pseudogenes, and most differences are only of modest significance (*Figure 4E*). The differentially expressed genes are not linked to the *Sst1*$^S$ locus, but could derive from the partial low expression of *Sp140* and the loss of *Sp110* in B6.*Sst1*$^S$, as compared to the complete loss of functional SP140 protein and retention of SP110 in *Sp140*$^{-/-}$ mice. Alternatively, the minor differences in gene expression between B6.*Sst1*$^S$ and B6.*Sp140*$^{-/-}$ mice could arise from additional genetic background or microbiota differences between B6.*Sst1*$^S$ and *Sp140*$^{-/-}$ mice. Interestingly, the gene most significantly differentially expressed between B6.*Sst1*$^S$ and *Sp140*$^{-/-}$ mice (i.e., with the smallest adjusted p-value) was *Sp110* (*Figure 4E*). This result is expected, given that *Sp110* is not expressed in B6.*Sst1*$^S$ but is retained in

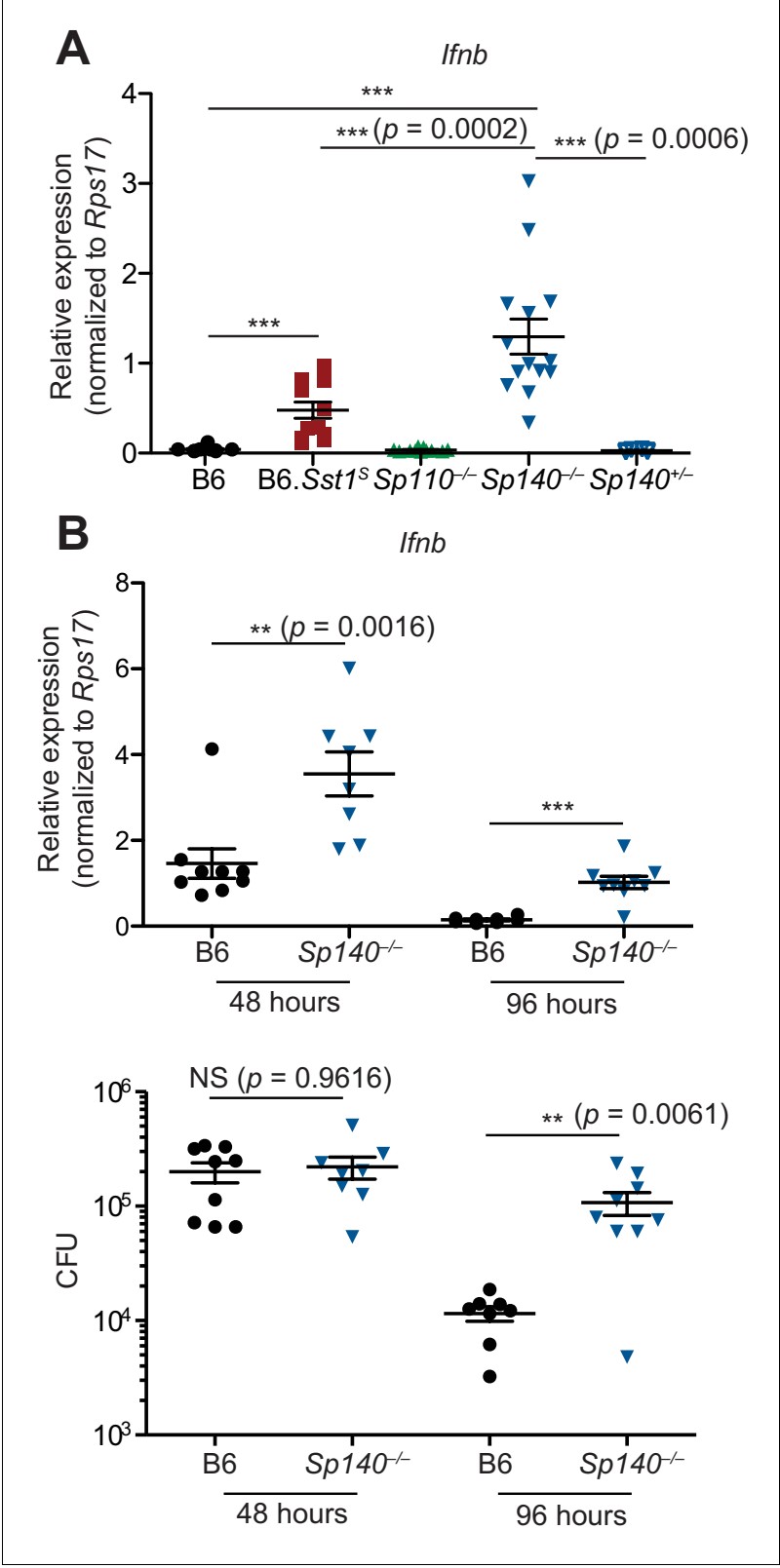

**Figure 3.** *Sp140⁻/⁻* mice have elevated *Ifnb* transcripts during bacterial infection. (A) Mice were infected with *Mycobacterium tuberculosis* and at 28 days post-infection, lungs were processed for total RNA, which was used for RT-qPCR. Combined results of two independent experiments. (B) Mice were infected with *Legionella pneumophila* and RT-qPCR (top panel) and CFU enumeration (bottom panel) was performed on lungs collected at indicated

*Figure 3 continued on next page*

*Figure 3 continued*

times. Combined results of two independent infections. All mice were bred in-house, $Sp140^{-/-}$ and $Sp140^{+/-}$ were littermates. (A, B) Mann-Whitney test. *p≤0.05; **p≤0.01; ***p≤0.005. CFU, colony-forming unit; RT-qPCR, real-time quantitative-polymerase chain reaction.
The online version of this article includes the following figure supplement(s) for figure 3:

**Figure supplement 1.** BMMs from B6.$Sst1^S$ and $Sp140^{-/-}$ mice show increased cell death upon stimulation with polyI:C, which is dependent upon IFNAR signaling.

our $Sp140^{-/-}$ mice (*Figure 2—figure supplement 1D*). Together, these results show that while they are not identical, the transcriptomes of $Sp140^{-/-}$ and B6.$Sst1^S$ mice greatly overlap during *M. tuberculosis* infection, and importantly, both strains exhibit a similar type I IFN signature. Given the susceptibility of B6.$Sst1^S$ mice is due to overproduction of type I IFN (*Ji et al., 2019*), we hypothesized that type I IFNs might also mediate the susceptibility of $Sp140^{-/-}$ mice.

## Susceptibility of $Sp140^{-/-}$ mice to bacterial infections depends on type I IFN signaling

To determine whether type I IFNs exacerbate *M. tuberculosis* infection of $Sp140^{-/-}$ mice, *M. tuberculosis*-infected $Sp140^{-/-}$ mice were treated with a blocking antibody against IFNAR1. Compared to mice that only received isotype control antibody, $Sp140^{-/-}$ mice that received the anti-IFNAR1 antibody had reduced bacterial burdens in their lungs (*Figure 5—figure supplement 1*). We also generated $Sp140^{-/-}$ $Ifnar^{-/-}$ double-deficient mice and infected them with *M. tuberculosis* (*Figure 5A-B*). Loss of *Ifnar* protected $Sp140^{-/-}$ mice from weight loss (*Figure 5A*) and reduced bacterial burdens at day 25 post-infection, similar to those seen in B6 mice (*Figure 5B*). Furthermore, $Sp140^{-/-}$ $Ifnar^{-/-}$ mice were partially protected from *L. pneumophila* infection, to a similar degree as B6.$Sst1^S$ $Ifnar^{-/-}$ mice (*Figure 5C-D*). These results show that similar to B6.$Sst1^S$ mice, type I IFN signaling is responsible for the susceptibility of $Sp140^{-/-}$ mice to *M. tuberculosis*, and partially responsible for the susceptibility of $Sp140^{-/-}$ mice to *L. pneumophila*.

## Discussion

Humans and other vertebrates encounter diverse classes of pathogens, including viruses, bacteria, fungi, and parasites. In response, vertebrate immune systems have evolved stereotypical responses appropriate for distinct pathogen types. For example, type I IFN-driven immunity is generally critical for defense against viruses (*Schneider et al., 2014*; *Stetson and Medzhitov, 2006*), whereas type II IFN (IFN-γ)-driven immunity mediates resistance to intracellular pathogens (*Crisler and Lenz, 2018*). Additionally, IL-1 is important for inducing neutrophil and other responses against extracellular pathogens (*Mantovani et al., 2019*), and IL-4/–13 (type 2 immunity) orchestrates responses to helminths and other parasites (*Locksley, 1994*). Thus, an important question is how the immune system generates responses that are appropriate for resistance to a specific pathogen while repressing inappropriate responses. The alternative strategy of making all types of responses to all pathogens appears not to be employed, possibly because it would be too energetically costly, or incur too much inflammatory damage to the host. Although there is still much to be learned, it appears that negative feedback is essential to enforce choices between possible types of immune responses. For example, IL-4 and IFN-γ have long been appreciated to act as reciprocal negative antagonists of each other (*Locksley, 1994*). In addition, anti-viral type I IFNs negatively regulate IFN-γ and IL-1-driven anti-bacterial responses via diverse mechanisms (*Donovan et al., 2017*; *Moreira-Teixeira et al., 2018*). Although negative regulation of IFN-γ/IL-1 by type I IFN is likely beneficial to limit immunopathology during viral infections, $Sst1^S$ mice provide an example of how excessive or inappropriate negative regulation by type I IFN can also be detrimental during bacterial infections (*He et al., 2013*; *Ji et al., 2019*). In this study, we therefore sought to understand the molecular basis by which wild-type ($Sst1^R$) mice are able to restrain type I IFNs appropriately during bacterial infections.

Although the *Sst1* locus was first described in 2005 (*Pan et al., 2005*), further genetic analysis of the locus has been hindered by its extreme repetitiveness and the concomitant difficulty in generating specific loss-of-function mutations in *Sst1*-linked genes. In particular, the loss of *Sp110 (Ipr1)* has

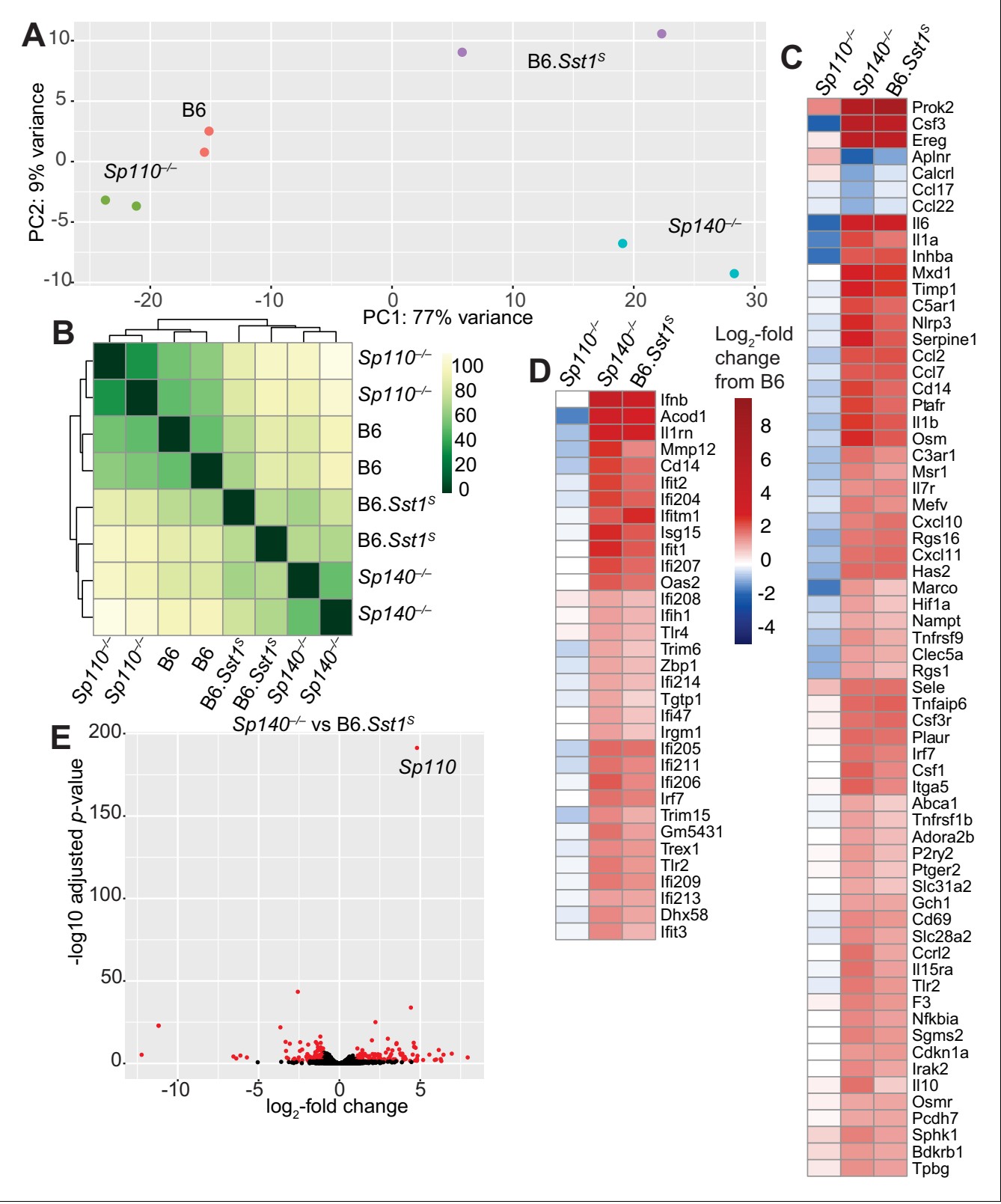

**Figure 4.** Global gene expression analysis of *Sp110⁻/⁻*, *Sp140⁻/⁻*, and B6.*Sst1ˢ* lungs after *Mycobacterium tuberculosis* infection. (A) PCA or (B) Euclidean distance analysis of all the samples. (C, D) Heatmaps of gene expression in log₂-fold change from *M. tuberculosis*-infected B6. Genes shown are those significantly different between *Sp140⁻/⁻* and B6. (C) GSEA Hallmark inflammatory response; and (D) GO type I IFN response genes. (E)
*Figure 4 continued on next page*

*Figure 4 continued*

Volcano plot comparing *Sp140*$^{-/-}$ to B6.*Sst1*$^{S}$ expression. Dots in red are twofold differentially expressed with adjusted p-value ≤0.05. PCA, principal component analysis.

The online version of this article includes the following figure supplement(s) for figure 4:

**Figure supplement 1.** B6.*Sst1*$^{S}$ and *Sp140*$^{-/-}$ lungs exhibit elevated transcript levels of the interferon-stimulated gene *Il1rn* during *Mycobacterium tuberculosis* infection.

**Figure supplement 2.** Expression of SP family members in *Sp140*$^{-/-}$ and *Sp110*$^{-/-}$ mouse lungs during *Mycobacterium tuberculosis* infection.

long been proposed to explain the susceptibility of *Sst1* mice to bacterial infections. However, while we could confirm the loss of *Sp110* expression in *Sst1*$^{S}$ mice, specific *Sp110*$^{-/-}$ mice were never generated and thus its essential role in host defense has been unclear. The advent of CRISPR/Cas9-

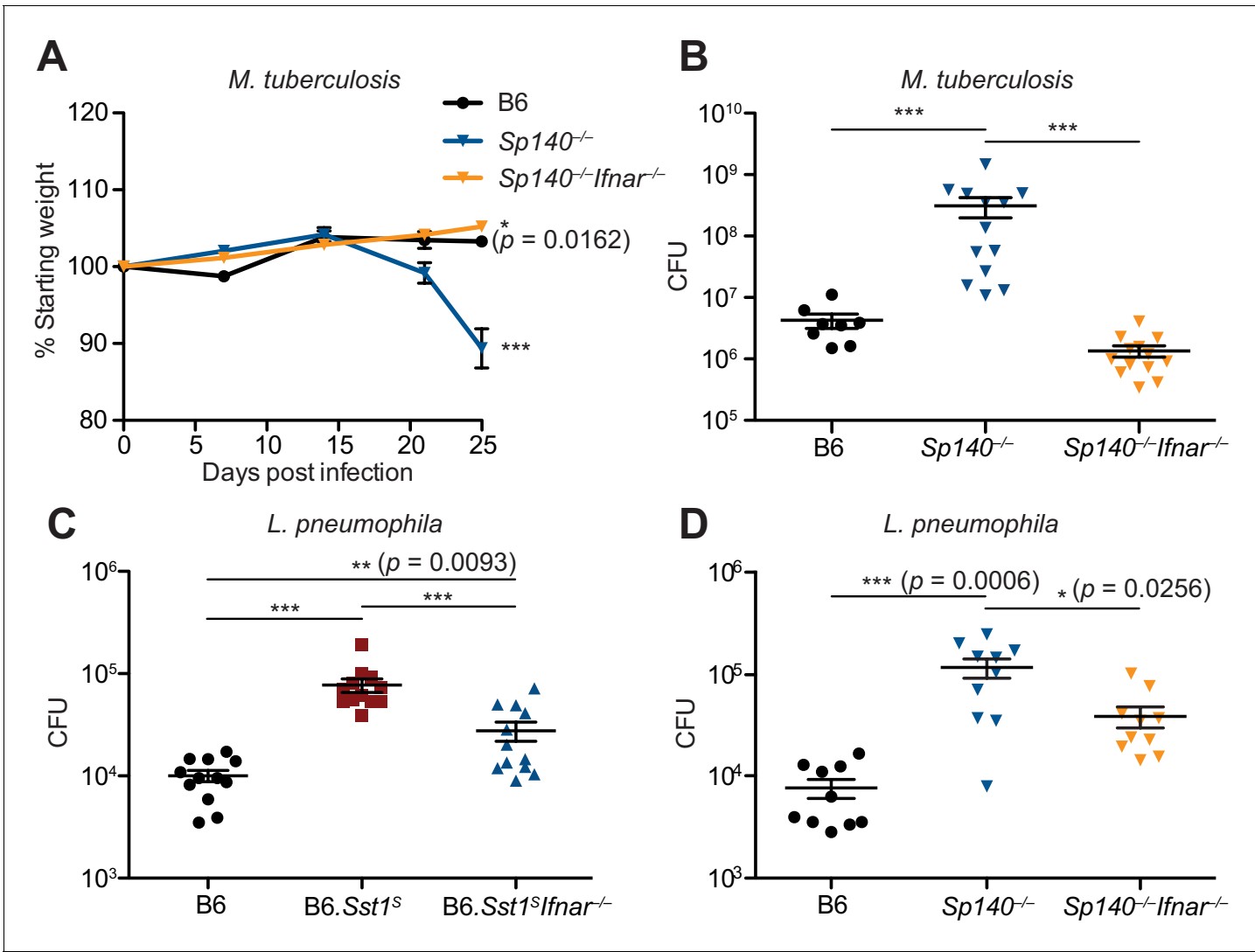

**Figure 5.** Susceptibility of *Sp140*$^{-/-}$ to *Mycobacterium tuberculosis* and *Legionella pneumophila* is dependent on type I IFN signaling. (A, B) Mice were infected with *M. tuberculosis* and measured for (A) body weight, and (B) bacterial burdens at day 25. Statistics in (A) show comparison to B6; data are from 9 B6, 13 *Sp140*$^{-/-}$, and 13 *Sp140*$^{-/-}$ *Ifnar*$^{-/-}$ mice. Combined results of two experiments. (C, D) Bacteria burden in *L. pneumophila*-infected mice at 96 hr. Combined results of two experiments. All mice were bred in-house (A, B, D); all but B6 were bred in-house (C). Mann-Whitney test (A–D). *p≤0.05; **p≤0.01; ***p≤0.005.

The online version of this article includes the following figure supplement(s) for figure 5:

**Figure supplement 1.** Antibody blockade of IFNAR1 reduces bacterial burden in *Sp140*$^{-/-}$ mice during *Mycobacterium tuberculosis* infection.

based methods of genome engineering allowed us to generate *Sp110*$^{-/-}$ mice. Unexpectedly, we found that *Sp110*$^{-/-}$ mice were fully resistant to *M. tuberculosis* infection, and we thus conclude that lack of *Sp110* is not sufficient to explain the *Sst1*$^S$ phenotype. An important caveat of genetic studies of the *Sst1* locus is that generating specific gene knockouts is still nearly impossible in this genetic region, even with CRISPR/Cas9. Indeed, the guide sequence used to target exon 3 of *Sp110* also targets an unknown number of pseudogene copies of *Sp110*-like genes located within the unassembled adjacent 'homogenously staining region' of mouse chromosome 1. Thus, we expect that additional off-target mutations are likely present in our *Sp110*$^{-/-}$ mutant mice. However, given that the *Sp110* pseudogenes are not known to be expressed, we consider it unlikely that collateral mutations would affect our conclusions. Moreover, any off-target mutations should differ among the three founder mice we analyzed and are thus unlikely to explain the consistent resistant phenotype we observed in all three founders. Additionally, we did not observe major changes in gene expression for other SP family members (*Sp100* and *Sp140*) in *Sp110*$^{-/-}$ mice during *M. tuberculosis* infection (**Figure 4—figure supplement 2**). Finally, since we were able to establish that all the founders at a minimum lack SP110 protein, additional mutations would not affect our conclusion that *Sp110* is not essential for resistance to *M. tuberculosis*.

Given that loss of *Sp110* was not sufficient to explain the susceptibility of *Sst1*$^S$ mice to bacterial infections, we considered other explanations. We found that *Sst1*$^S$ mice also lack expression of *Sp140*, an *Sst1*-linked homolog of *Sp110*. Our data suggest that deletion of *Sp140* is sufficient to recapitulate the full *Sst1*$^S$ phenotype including broad susceptibility to multiple bacterial infections, including *M. tuberculosis* and *L. pneumophila*. From the analysis of RNA-seq data generated from *M tuberculosis*-infected lungs, we found that the transcriptomes of *Sp140*$^{-/-}$ and B6.*Sst1*$^S$ greatly overlap and display an elevated ISG signature. The elevated production of *Ifnb* mRNA and *Il1rn* mRNA seen by RNAseq was validated by RT-qPCR. Enhanced *Ifnb* production and *Ifnar*-dependent cell death were also observed during in vitro experiments with BMMs. A causative role for type I IFNs in the phenotype of *Sp140*$^{-/-}$ and *Sst1*$^S$ mice was seen in the reduced susceptibility of *Sp140*$^{-/-}$ *Ifnar*$^{-/-}$ and B6.*Sst1*$^S$ *Ifnar*$^{-/-}$ mice to bacterial infection. Overall, we therefore conclude that loss of *Sp140* likely explains the *Sst1*-linked hyper type I IFN-driven susceptibility to bacterial infections. It remains possible that the additional loss of *Sp110* in *Sst1*$^S$ mice further exacerbates the *Sst1*$^S$ susceptibility phenotype as compared to *Sp140*$^{-/-}$ mice. However, in our studies, we did not observe a consistent difference in susceptibility between *Sst1*$^S$ (i.e., *Sp110*$^{-/-}$ *Sp140*$^{-/-}$) mice as compared to our *Sp140*$^{-/-}$ mice.

Another important caveat to our study is that it remains possible that our *Sp140*$^{-/-}$ mice carry additional mutations that contribute to, or even fully explain, their observed phenotype. This concern is somewhat ameliorated by our analysis of two independent *Sp140*$^{-/-}$ founders, both of which exhibited susceptibility to *M. tuberculosis* (**Figure 2—figure supplement 1E**). We confirmed there is normal SP110 protein levels in bone marrow macrophages from *Sp140*$^{-/-}$ mice (**Figure 2—figure supplement 1D**), and normal levels of *Sp110* and *Sp100* mRNA in the lungs of *M. tuberculosis*-infected *Sp140*$^{-/-}$ mice (**Figure 4—figure supplement 2**). Thus, collateral loss of SP100 or SP110 is unlikely to explain the phenotype of our *Sp140*$^{-/-}$ mice. To address the possibility of mutations in unannotated SP-like genes, we used deep amplicon sequencing of genomic DNA and cDNA from *Sp140*$^{-/-}$ mice. We confirmed that both founder lines harbored distinct off-target mutations. Most of the identified off-target mutations are in previously unidentified sequences that likely originate from *Sp140* paralogs within the unmapped HSR. Most HSR-linked paralogs are believed to be pseudogenes, and indeed, the off-target mutated genes appear to be expressed at a far lower level than *Sp140* in lungs during *M. tuberculosis* infection. In one of our *Sp140*$^{-/-}$ lines, we identified an off-target mutated *Sp140*-like paralog that was expressed at detectable levels in the lungs of *M. tuberculosis*-infected mice. This paralog was 100% identical to *Sp140* in the sequenced region and was only distinguished from *Sp140* itself because it lacked the deletion that was introduced into the edited *Sp140* gene. Importantly, this previously undescribed *Sp140*-like expressed sequence was not mutated in our second *Sp140*$^{-/-}$ line and is thus unlikely to explain resistance to *M. tuberculosis* infection. As an alternative approach to confirm the phenotype of *Sp140*$^{-/-}$ mice is due to loss of *Sp140*, we overexpressed *Sp140* in *Sp140*$^{-/-}$ BMMs. Crucially, we found *Sp140* complements aberrant elevated *Ifnb* transcription exhibited by *Sp140*$^{-/-}$ BMMs upon TNFα stimulation. Finally, *Sp110* and *Sp140* are the only two *Sst1*-linked genes that we were able to find to be differentially expressed between B6 and B6.*Sst1*$^S$ mice, and as discussed above, our genetic studies suggest little

role for the loss of *Sp110*. Thus, while it is formally possible that an edited *Sp140* homolog that was not identified by our amplicon sequencing contributes to the susceptibility to bacterial infection and elevated type I IFN in *Sp140*$^{-/-}$ mice, the most parsimonious explanation of our data is that deficiency in *Sp140* accounts for the *Sst1*$^S$ phenotype. We expect that future mechanistic studies will be critical to further confirm this conclusion.

Because *Sp140* is inducible by IFN-γ, our results suggest the existence of a novel feedback loop by which IFN-γ acts to repress the transcription of type I IFNs via SP140. This feedback loop appears to be essential for host defense against diverse bacterial pathogens. A major question that remains is how SP140 acts to repress the type I IFN response. SP140 contains DNA/chromatin-binding domains, such as SAND, PHD, and BRDs, which suggest the hypothesis that SP140 functions as a direct transcriptional repressor of type I IFN genes. However, much more indirect mechanisms are also possible. Recent studies suggest that hyper type I IFN responses in TNF-stimulated B6.*Sst1*$^S$ BMMs derive from aberrant oxidative stress that activates the kinase JNK and ultimately results in a non-resolving stress response that promotes necrosis (*Bhattacharya et al., 2021*; *Brownhill et al., 2020*). Interestingly, mouse SP140 localizes to nuclear structures called PML bodies. PML bodies are implicated in a variety of cell processes, such as apoptosis, cell cycle, DNA damage response, senescence, and cell-intrinsic anti-viral responses (*Scherer and Stamminger, 2016*). Whether or not the repressive effects of SP140 on type I IFN expression occur via the activity of PML bodies is an important outstanding question. Another major question is whether or how the repression of type I IFNs by SP140 is specific for bacterial infections and, if not, whether the presence of SP140 impairs anti-viral immunity. Finally, polymorphisms in human SP140 are associated with chronic lymphocytic leukemia, Crohn's disease, and multiple sclerosis (*Franke et al., 2010*; *International IBD Genetics Consortium (IIBDGC) et al., 2012*; *Karaky et al., 2018*; *Matesanz et al., 2015*; *Slager et al., 2013*). Studies using siRNA and shRNA-mediated knockdown have also implicated SP140 in the repression of lineage-inappropriate genes in macrophages (*Mehta et al., 2017*). Our generation of *Sp140*$^{-/-}$ mice is therefore important to permit future studies into these alternative roles of SP140.

# Materials and methods

## Key resources table

| Reagent type (species) or resource | Designation | Source or reference | Identifiers | Additional information |
|---|---|---|---|---|
| Gene (*Mus musculus*) | *Sp110* | GenBank | Gene ID: 109032 | |
| Gene (*Mus musculus*) | *Sp140* | GenBank | Gene ID: 434484 | |
| Strain, strain background (*M. tuberculosis*, Erdman) | *M. tuberculosis* | Sarah Stanley, University of California, Berkeley | Erdman | |
| Strain, strain background (*Legionella pneumophila*, JR32 ΔflaA) | *L. pneumophila* | Dario Zamboni, University of São Paulo, Brazil | JR32 | |
| Genetic reagent (*Mus musculus*) | *Sp110*$^{-/-}$ | This paper | | (C57BL/6J background) |
| Genetic reagent (*Mus musculus*) | *Sp140*$^{-/-}$ | This paper | | (C57BL/6J background) |
| Genetic reagent (*Mus musculus*) | B6.129S2-*Ifnar1*$^{tm1Agt}$/Mmjax | Jackson Laboratory | RRID:MMRRC_032045-JAX | |
| Genetic reagent (*Mus musculus*) | B6J.C3-*Sst*$^{C3HeB/FeJ}$Krmn | Igor Kramnik, Boston University | | |
| Cell line (*Homo sapiens*) | GP-2 293 | UC Berkeley Cell culture Facility | RRID:CVCL_WI48 | |

*Continued on next page*

*Continued*

| Reagent type (species) or resource | Designation | Source or reference | Identifiers | Additional information |
|---|---|---|---|---|
| Antibody | Rabbit polyclonal anti-mouse SP110 (serum) | Covance, this paper | | WB (1:1000) |
| Antibody | Rabbit polyclonal anti-mouse SP140 (serum) | Covance, this paper | | WB (1:1000) |
| Antibody | Mouse monoclonal anti-mouse SP110 (hybridoma) | Igor Kramnik, Boston University | | WB (1:1000) |
| Antibody | Mouse anti-human IFNGR-α chain (isotype control) | Leinco Technologies, Inc | Cat #: GIR208 | Mouse injection (500 µg) |
| Antibody | Mouse anti-mouse IFNAR1 | Leinco Technologies, Inc | Cat #: MAR1-5A3 | Mouse injection (500 µg) |
| Recombinant DNA reagent | SINV-mincmvSp140-pgkAmetrine (plasmid) | This paper | | Derived from pTMGP vector (Addgene plasmid # 32716, RRID:Addgene_32716) |
| Recombinant DNA reagent | SINV-Gal4-mincmv-mNeonGreen-pgkAmetrine (plasmid) | This paper | | Derived from pTMGP vector (Addgene plasmid # 32716, RRID:Addgene_32716) |
| Recombinant DNA reagent | pMD2.G | Addgene | RRID:Addgene_12259 plasmid #32716 | |
| Peptide, recombinant protein | Recombinant murine TNF alpha | R&D Systems | Cat #: 410-TRNC-010 | BMM stimulation (10 ng/mL) |
| Peptide, recombinant protein | Recombinant murine interferon gamma | Biolegend | Cat #: 575304 | BMM stimulation (5–10 ng/mL) |
| Peptide, recombinant protein | Retronectin | Takara | T100 | |
| Sequence-based reagent | Sp110 fwd | This paper | Genotyping primers (*Sp110*) | CTCTCCGCTCGGTGACTAC |
| Sequence-based reagent | Sp110 rev | This paper | Genotyping primers (*Sp110*) | CTGCACATGTGACAAGGATCTC |
| Sequence-based reagent | Sp140-1 fwd | This paper | Genotyping primers (*Sp140*) | ACGAATAGCAAGCAGGAATGCT |
| Sequence-based reagent | Sp140-1 rev | This paper | Genotyping primers (*Sp140*) | GGTTCCGGCTGAGCACTTAT |
| Sequence-based reagent | Sp140-2 fwd | This paper | Genotyping primers (*Sp140*) | TGAGGACAGAACTCAGGGAG |
| Sequence-based reagent | Sp140-2 rev | This paper | Genotyping primers (*Sp140*) | ACACGCCTTTAATCCCAGCATTT |
| Sequence-based reagent | *Ifnb* sense | This paper | RT-qPCR primers (*Ifnb*) | GTCCTCAACTGCTCTCCACT |
| Sequence-based reagent | *Ifnb* antisense | This paper | RT-qPCR primers (*Ifnb*) | CCTGCAACCACCACTCATTC |
| Commercial assay or kit | E.Z.N.A. Total RNA Kit I | Omega Biotek | Cat #: R6834-02 | |
| Chemical compound, drug | polyI:C | Invivogen | Cat #: tlrl-picw | BMM stimulation (100 µg/mL) |

## Mice

All mice were specific pathogen-free, maintained under a 12 hr light-dark cycle (7 a.m. to 7 p.m.), and given a standard chow diet (Harlan irradiated laboratory animal diet) ad libitum. All mice were sex- and age-matched at 6–10 weeks old at the beginning of infections. Littermates were used as indicated in the figure legends. C57BL/6J (B6) and B6.129S2-$Ifnar1^{tm1Agt}$/Mmjax ($Ifnar^{-/-}$) were

originally purchased from Jackson Laboratories and subsequently bred at UC Berkeley. B6J.C3-$Sst^{C3HeB/FeJ}$Krmn mice (referred to as B6.$Sst1^S$ throughout) were from the colony of I. Kramnik at Boston University and then transferred to UC Berkeley. CRISPR/Cas9 targeting was performed by pronuclear injection of Cas9 mRNA and sgRNA into fertilized zygotes from colony-born C57BL/6J mice, essentially as described previously (*Wang et al., 2013*). Founder mice were genotyped as described below, and founders carrying *Sp140* mutations were bred one generation to C57BL/6J to separate modified *Sp140* haplotypes. Homozygous lines were generated by interbreeding heterozygotes carrying matched *Sp140* haplotypes. $Sp140^{-/-}$ $Ifnar^{-/-}$ mice were generated by crossing the $Sp140^{-/-}$ and $Ifnar^{-/-}$ mice in-house. All animals used in experiments were bred in-house unless otherwise noted in the figure legends. All animal experiments complied with the regulatory standards of, and were approved by, the University of California Berkeley Institutional Animal Care and Use Committee.

## Genotyping of *Sp110* alleles

Exon 3 and the surrounding intronic regions were amplified by PCR using the following primers (all 5′–3′): Sp110 fwd, CTCTCCGCTCGGTGACTAC, and rev, CTGCACATGTGACAAGGATCTC. The primer combinations were designed to distinguish *Sp110* from other *Sp110*-like genes. Primers were used at 200 nM in each 20 µl reaction with 1× Dreamtaq Green PCR Master Mix (Thermo Fisher Scientific). Cleaned PCR products were diluted at 1:10 and sequenced using Sanger sequencing (Elim Biopharm).

## Genotyping of *Sp140* alleles

Exon 3 and the surrounding intronic regions were amplified by bracket PCR using the following primers (all 5′–3′): Sp140-1 fwd, ACGAATAGCAAGCAGGAATGCT, and rev, GGTTCCGGCTGAGCAC TTAT. The PCR products are diluted at 1:10 and 2 µl were used as template for the second PCR using the following primers: Sp140-2 fwd, TGAGGACAGAACTCAGGGAG, and rev, ACACGCC TTTAATCCCAGCATTT. The primer combinations were designed to distinguish *Sp140* from other *Sp140*-like genes. Primers were used at 200 nM in each 20 µl reaction with 1× Dreamtaq Green PCR Master Mix (Thermo Fisher Scientific). Cleaned PCR products were diluted at 1:10 and sequenced using Sanger sequencing (Elim Biopharm). PCRs were performed as described above for *Sp110* and sequenced using Sanger sequencing (Elim Biopharm).

## *Mycobacterium tuberculosis* infections

*M. tuberculosis* strain Erdman (gift of S.A. Stanley) was used for all infections. Frozen stocks of this wild-type strain were made from a single culture and used for all experiments. Cultures for infection were grown in Middlebrook 7H9 liquid medium supplemented with 10% albumin-dextrose-saline, 0.4% glycerol, and 0.05% Tween-80 for 5 days at 37°C. Mice were aerosol infected using an inhalation exposure system (Glas-Col, Terre Haute, IN). A total of 9 ml of diluted culture was loaded into the nebulizer calibrated to deliver ~20–50 bacteria per mouse as confirmed by measurement of CFUs in the lungs 1 day following infection. Mice were sacrificed at various days post-infection (as described in figure legends) to measure CFUs and RNA levels. All but one lung lobe was homogenized in phosphate-buffered saline (PBS) plus 0.05% Tween-80, and serial dilutions were plated on 7H11 plates supplemented with 10% oleic acid, albumin, dextrose, and catalase and 0.5% glycerol. CFUs were counted 21 days after plating. The remaining lobe was used for histology or for RNA extraction. For histology, the sample was fixed in 10% formalin for at least 48 hr then stored in 70% ethanol. Samples were sent to Histowiz Inc for embedding in wax, sectioning and staining with H&E. For histologic grading, slides were scanned at 20× magnification and evaluated by a trained pathologist (Stephen L. Nishimura) for the extent of macrophage, lymphoid, and granulocytic infiltration. The extent of infiltration was graded on a 0–4 scale with 0 being the least and four being the greatest. The extent of necrosis was similarly estimated. For survival experiments, mice were monitored for weight loss and were euthanized when they reached a humane endpoint as determined by the University of California Berkeley Institutional Animal Care and Use Committee.

## *Legionella pneumophila* infections

Infections were performed using *L. pneumophila* strain JR32 Δ*flaA* (from the lab of D.S. Zamboni) as previously described (Gonçalves et al., 2019; Mascarenhas et al., 2015). Briefly, frozen cultures were streaked out on to BCYE plates to obtain single colonies. A single colony was chosen and streaked on to a new BCYE plate to obtain a 1 cm by 1 cm square bacterial lawn, and incubated for 2 days at 37°C. The patch was solubilized in autoclaved MilliQ water and the optical density was measured at 600 nm. Culture was diluted to $2.5 \times 10^6$ bacteria/ml in sterile PBS. The mice were first anesthetized with ketamine and xylazine (90 mg/kg and 5 mg/kg, respectively) by intraperitoneal injection then infected intranasally with 40 µL with PBS containing a final dilution of $1 \times 10^5$ bacteria per mouse. For enumerating CFUs, the lungs were harvested and homogenized in 5 mL of autoclaved MilliQ water for 30 s, using a tissue homogenizer. Lung homogenates were diluted in autoclaved MilliQ water and plated on BCYE agar plates. CFUs were enumerated after plates were incubated for 4 days at 37°C.

## Bone marrow-derived macrophages and TNF treatment

Bone marrow was harvested from mouse femurs and tibias, and cells were differentiated by culture on non-tissue culture-treated plates in RPMI supplemented with supernatant from 3T3-MCSF cells (gift of B. Beutler), 10% fetal bovine serum (FBS) (Gibco, CAT#16140071, LOT#1447825), 2 mM glutamine, 100 U/ml streptomycin, and 100 µg/ml penicillin in a humidified incubator (37°C, 5% $CO_2$). BMMs were harvested 6 days after plating and frozen in 95% FBS and 5% DMSO. For in vitro experiments, BMMs were thawed into media as described above for 4 hr in a humidified 37°C incubator. Adherent cells were washed with PBS, counted and replated at $1.2 \times 10^6$–$1.5 \times 10^6$ cells/well in a TC-treated six-well plate. Cells were treated with 10 ng/ml recombinant mouse TNFα (410-TRNC-010, R&D Systems) diluted in the media as described above.

## Quantitative/conventional RT-PCR

Total RNA from BMMs was extracted using E.Z.N.A. Total RNA Kit I (Omega Bio-tek) according to manufacturer's specifications. Total RNA from infected tissues was extracted by homogenizing in TRIzol reagent (Life Technologies) then mixing thoroughly with chloroform, both done under BSL3 conditions. Samples were then removed from the BSL3 facility and transferred to fresh tubes under BSL2 conditions. Aqueous phase was separated by centrifugation and RNA was further purified using the E.Z.N.A. Total RNA Kit I (Omega Bio-tek). Equal amounts of RNA from each sample were treated with DNase (RQ1, Promega) and cDNA was made using Superscript III (Invitrogen). Complementary cDNA reactions were primed with poly(dT) for the measurement of mature transcripts. For experiments with multiple time points, macrophage samples were frozen in the RLT buffer (Qiagen) and infected tissue samples in RNA*later* solution (Invitrogen) and processed to RNA at the same time. Quantitative PCR was performed using QuantiStudio 5 Real-Time PCR System (Applied Biosystems) with Power Sybr Green PCR Master Mix (Thermo Fisher Scientific) according to manufacturer's specifications. Transcript levels were normalized to housekeeping genes *Rps17*, *Actb*, and *Oaz1* unless otherwise specified. The following primers were used in this study. *Rps17* sense: CGCCATTA TCCCCAGCAAG; *Rps17* antisense: TGTCGGGATCCACCTCAATG; *Oaz1* sense: GTGGTGGCCTC TACATCGAG; *Oaz1* antisense: AGCAGATGAAAA CGTGGTCAG; *Actb* sense: CGCAGCCACTG TCGAGTC; *Actb* antisense: CCTTCTGACCCATTCCCACC; *Ifnb* sense: GTCCTCAACTGCTCTCCAC T; *Ifnb* antisense: CCTGCAACCACCACTCATTC; *Gbp4* sense: TGAGTACCTGGAGAATGCCCT; *Gbp4* antisense: TGGCCGAATTGGATGCTTGG; *Ifit3* sense: AGCCCACACCCAGCTTTT; *Ifit3* antisense: CAGAGATTCCCGGTTGACCT; *Tnfa* sense: TCTTCTCATTCCTGCTTGTGG; and *Tnfa* antisense: GGTCTGGGCCATAGAACTGA. Conventional RT-PCR shown in *Figure 2A* used the following primers. Sense: GTCCCTTGGAGTCTGTGTAGG; antisense: CATCCTGGGGCTCTTGTCTTG.

## Immunoblot

Samples were lysed in RIPA buffer with protease inhibitor cocktail (Roche) to obtain total protein lysate and were clarified by spinning at ~16,000×*g* for 30 min at 4°C. Clarified lysates were analyzed with Pierce BCA Protein Assay Kit (Thermo Fisher Scientific) according to the manufacturer's specification and diluted to the same concentration and denatured with SDS-loading buffer. Samples were separated on NuPAGE Bis–Tris 4–12% gradient gels (Thermo Fisher Scientific) following the

manufacturer's protocol. Gels were transferred onto ImmobilonFL PVDF membranes at 35 V for 90 min and blocked with Odyssey blocking buffer (Li-Cor). Proteins were detected on a Li-Cor Odyssey Blot Imager using the following primary and secondary antibodies. Rabbit anti-SP110 or SP140 serums were produced by Covance and used at 1:1000 dilution. Hybridoma cells expressing monoclonal anti-SP110 antibody were from the lab of I. Kramnik. Antibodies were produced in-house as previously described (*Ji et al., 2019*) and used at 100 ng/mL. Alexa Fluor 680-conjugated secondary antibodies (Invitrogen) were used at 0.4 mg/mL.

## RNA sequencing and analysis

Total RNA was isolated as described above. Illumina-compatible libraries were generated by the University of California, Berkeley, QB3 Vincent J. Coates Genomics Sequencing Laboratory. PolyA selection was performed to deplete rRNA. Libraries were constructed using Kapa Biosystem library preparation kits. The libraries were multiplexed and sequenced using one flow cell on Novaseq 6000 (Illumina) as 50 bp paired-end reads. Base calling was performed using bcl2fastq2 v2.20. The sequences were aligned to mm10 genome using Kallisto v.0.46.0 using standard parameters (*Pimentel et al., 2017*) and analyzed using Deseq2 (*Love et al., 2014*) and DEVis packages (*Price et al., 2019*). For Deseq2 and DEVis analysis, all raw counts were incremented by one to avoid excluding genes due to division by 0 in the normalization process, except for data shown in *Figure 4—figure supplement 2*. Fold changes were calculated with the apeglm shrinkage estimator (*Zhu et al., 2019*).

## Antibody-mediated neutralization

Mice were given anti-IFNAR1 antibody or isotype control once every 2 days, starting 7 days post-infection. All treatments were delivered by intraperitoneal injection. Mouse anti-mouse IFNAR1 (MAR1-5A3) and isotype control (GIR208, mouse anti-human IFNGR-α chain) were purchased from Leinco Technologies Inc. For injections, antibody stocks were diluted in sterile PBS and each mouse received 500 µg per injection.

## Amplicon sequencing and analysis

Amplicons comprising the 5′ intron of exon 3 of *Sp140* and the end of exon 3 were amplified from crude DNA from ear clips of B6 and *Sp140⁻/⁻* founder line 1 mice (sense: TCATATAACCCATAAA TCCATCATGACA; antisense: CCATTTAGGAAGAAGTGTTTTAGAGTCT) with PrimeStar PCR components (Takara, R010b) for 18 cycles according to manufacturer's specifications, then diluted 50-fold and barcoded for an additional 18 cycles with Illumina-compatible sequencing adaptors. Amplicons of *Sp140* exon 3 (sense: AATATCAAGAAACATGTAAGAACCTGGT; antisense: CCATTTAG-GAAGAAGTGTTTTAGAGTCT) and exons 2–3 (sense: GCAGAAGTTTCAGGAATATCAAGAAACATG TAAG; antisense: ACTTCTTCTGTACATTGCTGAGGATGT) were amplified from cDNA generated from lungs of B6, andboth lines of *Sp140⁻/⁻* mice infected with *M. tuberculosis* for 25 cycles with PrimeStar before barcoding. Libraries were generated by the University of California, Berkeley, QB3 Vincent J. Coates Genomics Sequencing Laboratory, and were multiplexed and sequenced on an Illumina Miseq platform with v2 chemistry and 300 bp single-end reads for DNA amplicons, and Illumina Miseq Nano platform with v3 chemistry for 300 bp single-end reads for cDNA amplicons. Reads were aligned with Burrows-Wheeler Aligner (BWA-MEM) with default parameters (*Li, 2013*; *Li and Durbin, 2009*) to chromosome one and non-localized genome contigs of the *Mus musculus* genome (assembly mm10) as well as the *Sp140* gene and transcript X1 (XM_030255396.1), converted to BAM files with samtools (*Li et al., 2009*), and visualized in IGV 2.8. Subsets of reads were extracted from alignment files using the Seqkit toolkit (*Shen et al., 2016*).

## Retroviral transduction of BMMs

Self-inactivating pTMGP vector (SINV) with either a minimal CMV promoter driving *Sp140* or a minimal CMV promoter and 4 Gal4 binding sites driving mNeonGreen, and the reporter mAmetrine driven by a PGK promoter, were cloned using Infusion (638910, Takara). pTGMP was from the lab of Scott Lowe (Addgene plasmid # 32716). Virus was harvested from GP-2 cells transfected with SINV vectors and VSV-G (pMD2.G, Addgene plasmid #12259) and grown in DMEM supplemented with 30% FBS and 2 mM glutamine, 100 U/ml streptomycin, and 100 µg/ml penicillin (adapted from

protocols described in *Schmidt et al., 2015*). Harvested virus was concentrated 100-fold by ultracentrifugation in RPMI before storage at −80°C. Virus was thawed and titrated on bone marrow to optimize transduction efficiency. Bone marrow was harvested as described above and the entirety of the bone marrow was plated in a non-TC 15 cm plate. The next day, bone marrow was harvested and transduced with SINV virus on plates coated with 10 µg/cm$^2$ Retronectin (T100, Takara) for 1.5–2 hr at 650×*g* and 37°C. After 2 days of additional culture, media was replenished, then transduced bone marrow was cultured for 3 additional days before sorting. Sorted transduced macrophages were stimulated with 5 ng/mL recombinant murine IFN-γ (575304, Biolegend) 12–14 hr before stimulation with 10 ng/mL recombinant TNFα as described above for 4 hr (FBS used in these experiments was from Omega, LOT 721017, CAT# FB-12). RNA isolation and RT-qPCR were performed as described above. No mycoplasma contamination was detected by PCR in GP-2 cells used for these experiments (sense: CACCATCTGTCACTCTGTTAACC; antisense: GGAGCAAACAGGATTAGATACCC), and GP-2 cells were authenticated by short tandem repeat DNA profiling by the UC Berkeley DNA Sequencing Facility.

## Quantification of cell death upon polyI:C treatment

Primary BMMs were derived from fresh or frozen bone marrow as described above for 6-7 days. BMMs were plated at 50,000–90,000 cells per well in 96-well non-TC treated plates, and stimulated for 16–24 hr with 100 µg/mL polyI:C (tlrl-picw, Invivogen). LDH assays were performed on supernatants after stimulation as previously described (*Decker and Lohmann-Matthes, 1988*). Similar results were obtained for BMMs cultured with FBS from Omega and Gibco (LOT 721017, CAT# FB-12, and LOT# 1447825, CAT#16140071, respectively).

## Statistical analysis

All data were analyzed with Mann-Whitney test unless otherwise noted. Tests were run using GraphPad Prism 5. *$p{\leq}0.05$; **$p{\leq}0.01$; ***$p{\leq}0.005$. All error bars are S.E. Figures show exact p-values for $p{>}0.0005$.

## Data accession

RNA-seq data is available at GEO, accession number GSE166114. Amplicon sequencing data is available at the SRA, BioProject accession number PRJNA698382.

# Acknowledgements

The authors thank the Stanley and Cox laboratories for support with *M. tuberculosis* experiments, L Flores, P Dietzen, and R Chavez for technical assistance, and members of the Vance, Barton, Cox, Stanley, and Portnoy labs for advice and discussions. Funding: REV is supported by an Investigator Award from the Howard Hughes Medical Institute. This work was also supported by NIH grants R37AI075039 and R01AI155634 (REV), P01AI066302 (REV and DAP), and R01HL134183 (SLN). REV and KHD were Burroughs Wellcome Fund Investigators in the Pathogenesis of Infectious Disease.

# Additional information

### Competing interests

Russell E Vance: consults for Ventus Therapeutics and is a Reviewing Editor for *eLife*. The other authors declare that no competing interests exist.

### Funding

| Funder | Grant reference number | Author |
| --- | --- | --- |
| National Institutes of Health | R37AI075039 | Russell E Vance |
| National Institutes of Health | R01AI155634 | Russell E Vance |
| Howard Hughes Medical Institute | Investigator Award | Russell E Vance |

| National Institutes of Health | P01AI066302 | Daniel A Portnoy
Russell E Vance |
| National Institutes of Health | R01HL134183 | Stephen L Nishimura |
| Burroughs Wellcome Fund | | Russell E Vance
K Heran Darwin |

The funders had no role in study design, data collection and interpretation, or the decision to submit the work for publication.

## Author contributions

Daisy X Ji, Conceptualization, Data curation, Formal analysis, Investigation, Methodology, Writing - original draft, Writing - review and editing; Kristen C Witt, Conceptualization, Data curation, Formal analysis, Validation, Investigation, Methodology, Writing - original draft, Writing - review and editing; Dmitri I Kotov, Validation, Investigation, Methodology, Writing - review and editing; Shally R Margolis, Investigation, Methodology, Writing - review and editing; Alexander Louie, Katherine J Chen, Harmandeep S Dhaliwal, Investigation; Victoria Chevée, Stephen L Nishimura, Validation, Investigation; Moritz M Gaidt, Investigation, Methodology; Angus Y Lee, Supervision, Investigation, Methodology; Dario S Zamboni, Supervision, Investigation, Writing - review and editing; Igor Kramnik, Conceptualization, Resources, Writing - review and editing; Daniel A Portnoy, Conceptualization, Resources, Funding acquisition, Project administration; K Heran Darwin, Conceptualization, Supervision, Methodology, Writing - review and editing; Russell E Vance, Conceptualization, Resources, Supervision, Funding acquisition, Writing - original draft, Project administration, Writing - review and editing

## Author ORCIDs

Daisy X Ji https://orcid.org/0000-0002-9148-3620
Kristen C Witt https://orcid.org/0000-0001-8744-9457
Dmitri I Kotov http://orcid.org/0000-0001-7843-1503
Dario S Zamboni http://orcid.org/0000-0002-7856-7512
Igor Kramnik http://orcid.org/0000-0001-6511-9246
Russell E Vance https://orcid.org/0000-0002-6686-3912

## Ethics

Animal experimentation: This study was performed in strict accordance with the recommendations in the Guide for the Care and Use of Laboratory Animals of the National Institutes of Health. Animal studies were approved by the UC Berkeley Animal Care and Use Committee (current protocol number: AUP-2014-09-6665-2).

## Decision letter and Author response

Decision letter https://doi.org/10.7554/eLife.67290.sa1
Author response https://doi.org/10.7554/eLife.67290.sa2

# Additional files

## Supplementary files

• Transparent reporting form

## Data availability

RNA-seq data is available at GEO, accession number GSE166114. Amplicon sequencing data is available at the SRA, BioProject accession number PRJNA698382.

The following datasets were generated:

| Author(s) | Year | Dataset title | Dataset URL | Database and Identifier |
|---|---|---|---|---|
| Ji D | 2021 | Role of the transcriptional regulator SP140 in resistance to bacterial infection via repression of type I interferons | https://www.ncbi.nlm.nih.gov/geo/query/acc.cgi?acc=GSE166114 | NCBI Gene Expression Omnibus, GSE166114 |
| Witt K | 2021 | Deep amplicon sequencing of Sp140-like genes in *Mus musculus* | https://www.ncbi.nlm.nih.gov/bioproject/?term=PRJNA698382 | NCBI BioProject, PRJNA698382 |

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
