## [Decision Letter]

**Acceptance summary:**

In this revised manuscript the authors' have addressed the reviewers' comments regarding this important and well-executed study into genetic factors that impact susceptibility to bacterial infections. Specifically, the authors identify the gene SP140 as the driving mutation resulting in susceptibility to a range of bacterial pathogens located within the severe susceptibility to tuberculosis (SST1) in mice. While our understanding of how SP140 functions remains unknown, the identification of a single gene within this locus that regulates Type I IFN is impactful and will be of broad interest for immunologists and microbiologists alike.

**Decision letter after peer review:**

Thank you for submitting your article "Role of the transcriptional regulator SP140 in resistance to bacterial infections via repression of type I interferons" for consideration by *eLife*. Your article has been reviewed by 3 peer reviewers, one of whom is a member of our Board of Reviewing Editors, and the evaluation has been overseen by Satyajit Rath as the Senior Editor. The reviewers have opted to remain anonymous.

Essential revisions:

The reviewers all agree that this is an important manuscript that would benefit from more thorough comparisons between Sp110-/-, Sp140-/-, and Sst1S mice. Specifically:

1) While the authors show many phenotypes of the Sst1S mouse are recapitulated in the Sp140-/- they did not fully compare them. For example, do the Sp140-/- mice/macrophages have increased necrosis and cell death during Mtb infection or following TNF stimulation? Are any of the lesions hypoxic? The pathology comparisons seem relatively superficial and could be quantified and examined more in depth to convincingly claim the effects of the Sst1S mice are recapitulated. In general, a more thorough histological analysis with more sections throughout the manuscript would be beneficial and aid in quantification. Particularly since this is a key feature of the Sst1S mice.

2) In Figure 3A, the Sp140-/- mice appear to drive even more type I IFN than Sst1 animals. Is this difference significant? How do the authors interpret these findings?

3) In Figure 2, the figure could be better labeled to distinguish A and B with RNA vs Protein. Additionally, the authors say "untreated nor IFNγ treated BMDMs produce SP140" but the data appear to only show IFNγ treated. This should clarified and/or the data shown.

4) The RNAseq experiments were performed with only two replicates, whereas at least 3 would be standard for these analyses.

5) The authors mention that they did not observe a consistent difference between the Sp140-/- and Sst1S mice, however, these strains are not completely overlapping in the expression profiling analysis. This could be because Sst1S has some expression of Sp140, whereas the KO has none, but this should be discussed. It would also be helpful to include WT B6 in the heatmaps in 4C and D.

6) In Figure 2, the levels of Sp140 protein in the Sp110-/- mice should be included in panel B to show that protein expression is intact (not just transcript) to rule out epistatic effects of missing both Sp140 and Sp110.

*Reviewer #1:*

In this manuscript, the authors build off their previous data where they have identified differences in the sst1 locus as responsible for differences in susceptibility of B6 and C3HeB/Fej mice to *Mycobacterium tuberculosis* infection. The authors have previously shown that this susceptibility is attributed to higher levels of type I IFN signaling and in particular, the ISG IL-1Ra. The sst1 locus contains many genes that could be contributing to the differential susceptibility in C3HeB/Fej mice, and the model in the field was that differences in Sp110 expression was a likely candidate to explain the susceptibility. However, in this manuscript, the authors show that it is not lower expression of Sp110, but instead decreased expression of another gene in the sst1 locus, Sp140, that contributes to the increased susceptibility of mice carrying the sst1S sequence to bacterial infections. This is a very significant and surprising finding, supported by very clear and convincing data from experiments performed with a high level of rigor. Although identification of the gene responsible for differences in susceptibility and outcomes during bacterial infections is an advance for the field, the manuscript stops there in terms of new insight and falls short of providing any additional information beyond what has already been published regarding how this gene or lucus is functioning to regulate immune responses to infection. This limited scope embodies the major concern for this otherwise strong manuscript.

*Reviewer #2:*

The authors have suggested the importance of SP140 for resistance to Mtb, Legionella infections in mice. They also provide evidence for IFNaR signalling in mediating the increased susceptibility of SP140-/- mice. While they attribute an important function of the transcriptional regulator SP140 to regulation of type I IFN responses by demonstrating the dysregulation of these responses in the SP140-/- mice, more direct evidence for this is needed.

*Reviewer #3:*

In this manuscript Ji et al. carefully examine candidate genes driving a previously described susceptibility within the severe susceptibility to tuberculosis (sst1). Surprisingly, mice deficient in the original candidate gene within this locus, SP110, showed no change in susceptibility to infection with *M. tuberculosis*. In contrast, the authors found that loss of a second gene in this locus, SP140, recapitulated many phenotypes seen in the SST1 mouse, including increased Type I IFN. SP140 susceptibility was reversed by blocking these exacerbated type I IFNs, similar to SST1 mice. RNAseq analysis identify changes in pro-inflammatory cytokines and type I IFNs. The strengths of this paper are the careful and controlled experiments to target and analyze mouse mutants within a notoriously challenging region with homopolymers. Their results are robust, convincing and will be of broad interest to the field of immunology and host-pathogen interactions. Convincingly identifying a single gene within this region that recapitulates many aspects of the SST1 mouse is very important. While a minor weakness is the lack of any mechanistic understanding of how SP140 functions, this is overcome by the impact of the other findings and it is anticipated that this mouse will now be a key resource to dissect the mechanisms of susceptibility in much greater detail.

---

## [Author Response]

Essential revisions:The reviewers all agree that this is an important manuscript that would benefit from more thorough comparisons between Sp110-/-, Sp140-/-, and Sst1S mice. Specifically:1) While the authors show many phenotypes of the Sst1S mouse are recapitulated in the Sp140-/- they did not fully compare them. For example, do the Sp140-/- mice/macrophages have increased necrosis and cell death during Mtb infection or following TNF stimulation? Are any of the lesions hypoxic?

The lesions observed in our histology samples are not hypoxic, since these samples are taken at a timepoint before hypoxia develops (Harper et al. (2012) PMC3266133). Importantly, we do not believe that hypoxia is the key to the susceptibility phenotype of these mice since we can see increased bacterial burdens before the mice reach the point at which the lesions become hypoxic. Our focus has been on the early immunological phenotypes that contribute to elevated CFU burdens. Others in the field may be interested in pursuing the hypoxia phenotype but this is not the focus of our paper. Hypoxia is also technically challenging to pursue since it only emerges at late timepoints. Similarly, we are not convinced that the cell death exhibited by B6.Sst1^S^ BMMs is driving the susceptibility of the mice. Indeed, we do not know how to specifically eliminate the cell death response in order to study its importance. For this reason, any discussion of the role of cell death would be purely speculative and thus our manuscript focuses on the hyper type I IFN production seen in both the B6.Sst1^S^ and Sp140^–/–^ mice. Because Ifnar deficiency rescues the susceptibility/CFU burdens of the mice, it is clear that type I IFNs are driving the disease phenotypes, and thus this is the focus of our manuscript. Nevertheless, to satisfy the reviewers, we have now included data in the revised manuscript showing that both B6.Sst1^S^ and Sp140^–/–^ BMMs die upon treatment with polyI:C (Figure 3 **−** figure supplement 1, lines 209-216). Importantly, this cell death response is reduced or eliminated by crosses to Ifnar^–/–^ mice, again emphasizing the primary importance of type I IFNs to the phenotype(s) of these mice.

We emphasize that we have fully compared CFUs and susceptibility for multiple bacterial infections in vivo, as well as transcriptional changes during *M. tuberculosis* infections in vivo, for both B6.Sst1^S^ and Sp140^–/–^ mice. These mice are essentially indistinguishable in terms of susceptibility to bacterial infection in vivo, and exhibit highly similar transcriptional responses to *M. tuberculosis* infection. As elaborated below, we have also expanded our comparison of lung pathology for B6.Sst1^S^ and Sp140^–/–^ mice. We feel that our studies establish the main claim of the manuscript, which is not that B6.Sst1^S^ and Sp140^–/–^ mice are identical, but rather, that loss of Sp140 is responsible for the susceptibility of Sst1^S^ mice to bacterial infections.

The pathology comparisons seem relatively superficial and could be quantified and examined more in depth to convincingly claim the effects of the Sst1S mice are recapitulated. In general, a more thorough histological analysis with more sections throughout the manuscript would be beneficial and aid in quantification. Particularly since this is a key feature of the Sst1S mice.

As argued above, we don’t necessarily believe that the histopathology seen in Sst1^S^ mice is the key to understanding the Sst1^S^ phenotype. Again, CFU burdens are clearly elevated in Sp140^–/–^ mice prior to the development of the highly organized/necrotic/hypoxic lesions, so we tend to think that the advanced features of these lesions are a late downstream consequence of a loss of bacterial control and is not therefore something we decided to focus on. Nevertheless, the point about better characterizing the histology is well taken. We have included a more in-depth analysis of the histology performed by a trained pathologist (newly added author Stephen Nishimura) and included scoring of macrophage, lymphoid, and granulocyte infiltration as well as extent of necrosis, in Figure 2 − figure supplement 2. We have also expanded our discussion of the histology in the paper (lines 165-173). As discussed above, at the relatively early timepoint analyzed (day 25), we do not observe dramatic differences in histology, but we do observe a trend of increased granulocyte infiltration for both B6.Sst1^S^ and Sp140^–/–^ lungs. Disease does appear to be more severe in the Sp140^–/–^ lungs as compared to Sst1^S^ lungs, but we do not wish to overemphasize this, as the differences are not statistically significant, and may be due to a small amount of residual Sp140 function in Sst1^S^ mice, or alternatively, due to microbiota differences between the strains. We have expanded our discussion to make these points. Again, our genetic data suggest that type I IFNs are the main driver of the Sst1^S^ phenotype and so we have focused our comparison to Sp140^–/–^ mice on this aspect of the phenotype.

2) In Figure 3A, the Sp140-/- mice appear to drive even more type I IFN than Sst1 animals. Is this difference significant? How do the authors interpret these findings?

This difference is significant and is now clearly marked in the figure and discussed further in the text (lines 219-224). As the reviewers have suggested, this difference could potentially derive from the partial low expression of Sp140 in B6.Sst1^S^, or possibly due to other differences (e.g., expression of Sp110, or microbiota differences) between the strains. We do not believe this difference is of major functional importance as the CFU at this timepoint is indistinguishable for B6.Sst1^S^ vs. Sp140^–/–^, there are no significant differences in weight loss or survival upon *M. tuberculosis* infection between B6.Sst1^S^ and Sp140^–/–^ mice, and the transcriptomes of both B6.Sst1^S^ and Sp140^–/–^ both show an elevated ISG signature upon *M. tuberculosis* infection. Furthermore, the differentially expressed genes between B6.Sst1^S^ and Sp140^–/–^ are not annotated as type 1 interferon stimulated genes, and we did not see statistically significant differences in expression for the interferon stimulated gene Il1rn genes between B6.Sst1^S^ and Sp140^–/–^ mice during *M. tuberculosis* infection (Figure 4 − figure supplement 1). We hypothesize that above a certain threshold, a moderate increase in type I IFN that we observe in Sp140^–/–^ mice as compared to B6.Sst1^S^ mice does not necessarily translate to further increased susceptibility to bacterial infection.

3) In Figure 2, the figure could be better labeled to distinguish A and B with RNA vs Protein. Additionally, the authors say "untreated nor IFNγ treated BMDMs produce SP140" but the data appear to only show IFNγ treated. This should clarified and/or the data shown.

The indicated figure and text have been modified for better clarity.

4) The RNAseq experiments were performed with only two replicates, whereas at least 3 would be standard for these analyses.

We recognize the limitations of RNAseq with 2 replicates. However, despite this limitation, our replicates show very similar patterns of gene expression and are sufficient to establish the broad claims we make in the manuscript. We were also able to observe significant differences in expression of numerous ISGs between B6, B6.Sst1^S^, and Sp140^–/–^ mice. Elevated type I interferon levels in Sp140^–/–^ mice during *M. tuberculosis* infection were confirmed by RT-qPCR (Figure 3A), and we confirmed the hypothesis that elevated type I interferon drove susceptibility of Sp140^–/–^ mice by infecting Sp140^–/–^ Ifnar^–/–^ mice. To further address the reviewer’s concern, we have also included new RT-qPCR data that confirms the elevated levels of the type 1 interferon stimulated gene Il1rn in our RNA-seq data for B6.Sst1^S^ and Sp140^–/–^ mice (Figure 4 − figure supplement 1). Because of batch effects, and experiment-to-experiment variability in in vivo lung infections, adding an additional replicate is not straightforward. We do not believe that additional RNAseq replicates would change the conclusions of our paper or even provide further significant insights into the phenotype.

5) The authors mention that they did not observe a consistent difference between the Sp140-/- and Sst1S mice, however, these strains are not completely overlapping in the expression profiling analysis. This could be because Sst1S has some expression of Sp140, whereas the KO has none, but this should be discussed. It would also be helpful to include WT B6 in the heatmaps in 4C and D.

We agree with the reviewers and have expanded our discussion of the transcriptional differences between B6.Sst1^S^ and Sp140^–/–^ mice in the text (lines 263-268). We do not expect B6.Sst1^S^ and Sp140^–/–^ mice to be transcriptionally identical, due to the loss of expression of Sp110 in B6.Sst1^S^ mice and possibly additional background or microbiota differences. However, as we state in our paper, our analysis shows that these mice show highly similar transcriptional responses to *M. tuberculosis* infection. The B6.Sst1^S^ and Sp140^–/–^ samples clearly cluster together along the PC1 axis in Figure 4A, which account for 77% of the variance in our RNA-seq data, and only separate for PC2, which accounts for only 9% of the variance in our data (lines 234-235). These samples also cluster together in Figure 4B, and in Figure 4E we show that differentially expressed genes between B6.Sst1^S^ and Sp140^–/–^ mice during *M. tuberculosis* infection are of relatively moderate significance, with the expected exception of Sp110. Again, our conclusion is not that Sp140^–/–^ mice are identical to B6.Sst1^s^ mice, but rather that Sp140 deficiency is responsible for the type I IFN-driven susceptibility to bacterial infection in B6.Sst1^S^ mice. The heatmaps in 4C and 4D display fold change relative to infected WT B6, which is now clarified in the figure legend, so showing this condition would not be informative.

6) In Figure 2, the levels of Sp140 protein in the Sp110-/- mice should be included in panel B to show that protein expression is intact (not just transcript) to rule out epistatic effects of missing both Sp140 and Sp110.

Although we agree that understanding the relative contributions of Sp110 and Sp140 is important, we do not understand the reasoning behind the experiment the reviewers suggest or what potential "epistatic effects" the reviewers have in mind. Note that we find that Sp110^–/–^ exhibit wild-type resistance to infection. If Sp140 protein expression is ablated in Sp110^–/–^ mice, we would predict the mice would phenocopy the susceptibility of B6.Sst1^S^ mice, which lack both Sp140 and Sp110. In other words, we fail to understand how Sp140 would be epistatic to Sp110 (i.e., how would loss of Sp140 obscure the phenotype of Sp110 deficiency?) since we know that loss of Sp140 alone (or in combination with loss of Sp110) results in susceptibility to infection. One interesting possibility might be that upregulation of Sp140 in Sp110^–/–^ mice could explain their resistance to bacterial infection. However, from our RNA-seq analysis, we observe very minor changes in transcript levels of Sp140 in our Sp110^–/–^ mice during *M. tuberculosis* infection relative to B6. We have added an additional supplement showing the expression of SP family members (Sp100, Sp110, Sp140) in both Sp110^–/–^ and Sp140^–/–^ mice (Figure 4—figure supplement 2; lines 249-258). We found that there is no significant difference in the levels of Sp100 in Sp110^–/–^ mice compared to B6 or B6.Sst1^S^. If anything, we observed a small decrease (rather than increase) in the expression of Sp140 in Sp110^–/–^ mice relative to B6 mice, but this change in Sp110^–/–^ mice is less than 2-fold and has a relatively high adjusted p-value (0.0343). Conversely, Sp140 is significantly upregulated in Sp110^–/–^ mice compared to B6.Sst1^S^ mice (fold change = 3.3849, adjusted p-value =1.49E-14 ). Overall, we do not observe major changes in the expression of other SP family members in Sp110^–/–^ and Sp140^–/–^ mice. Therefore, we believe that the targeting of these genes had specific rather than unanticipated epistatic effects.

In our view, however, the key experiment is to ensure that SP110 protein is present in Sp140^–/–^ mice, which are susceptible to bacterial infection. This experiment is important to rule out any effects of the loss of Sp140 on Sp110 expression, and to demonstrate that Sp110 is not sufficient to protect against bacterial infection. We show normal SP110 expression in Sp140^–/–^ macrophages in Figure 2—figure supplement 1D.

Reviewer #1:In this manuscript, the authors build off their previous data where they have identified differences in the sst1 locus as responsible for differences in susceptibility of B6 and C3HeB/Fej mice to Mycobacterium tuberculosis infection. The authors have previously shown that this susceptibility is attributed to higher levels of type I IFN signaling and in particular, the ISG IL-1Ra. The sst1 locus contains many genes that could be contributing to the differential susceptibility in C3HeB/Fej mice, and the model in the field was that differences in Sp110 expression was a likely candidate to explain the susceptibility. However, in this manuscript, the authors show that it is not lower expression of Sp110, but instead decreased expression of another gene in the sst1 locus, Sp140, that contributes to the increased susceptibility of mice carrying the sst1S sequence to bacterial infections. This is a very significant and surprising finding, supported by very clear and convincing data from experiments performed with a high level of rigor. Although identification of the gene responsible for differences in susceptibility and outcomes during bacterial infections is an advance for the field, the manuscript stops there in terms of new insight and falls short of providing any additional information beyond what has already been published regarding how this gene or lucus is functioning to regulate immune responses to infection. This limited scope embodies the major concern for this otherwise strong manuscript.

We thank for the reviewer for recognizing the importance of our discovery that loss of Sp140 (not Sp110) confers susceptibility to *M. tuberculosis*. Our generation of Sp140 deficient mice allows us to demonstrate, for the first time, that Sp140 is a negative regulator of type I IFNs. By generating crosses between Sp140^–/–^ and Ifnar^–/–^ mice, we further demonstrate that type I IFNs mediate the susceptibility of Sp140^–/–^ mice to *M. tuberculosis* and Legionella. The reviewer appears to believe that because IFNs were previously shown to mediate the phenotype of Sst1^S^ mice that somehow the function of Sp140 was already known. By contrast, we feel that in fact the function of Sp140 was not at all clear prior to our work, and that our work does indeed provide important mechanistic insight into the function of Sp140 as a regulator of type I IFNs. Sst1^S^ mice contain many genetic differences compared to B6 mice. It is only because of our work that we can now go back and reinterpret the prior work on Sst1^S^ mice, but this would not be possible without the work we have reported in this paper. Of course we would love to be able to describe more about the molecular mechanism by which Sp140 represses interferon transcription. This is indeed something we are working on. However, our preliminary experiments indicate this is not likely to be straightforward and will require considerable effort that is certainly beyond the scope of this current paper. It should be noted, for example, that Sp140 is in the same protein family as the well-known transcriptional regulator Aire. The mechanism by which Aire regulates gene expression has been studied for almost two decades and is still not entirely clear (and was certainly not clear in the initial foundational paper on Aire function published by Anderson et al. in Science in 2002). We expect the mechanism of Sp140 to be similarly complex. Importantly, we now know for the first time which protein to study mechanistically, i.e., SP140 instead of SP110.

Reviewer #2:The authors have suggested the importance of SP140 for resistance to Mtb, Legionella infections in mice. They also provide evidence for IFNaR signalling in mediating the increased susceptibility of SP140-/- mice. While they attribute an important function of the transcriptional regulator SP140 to regulation of type I IFN responses by demonstrating the dysregulation of these responses in the SP140-/- mice, more direct evidence for this is needed.

We appreciate the reviewer’s succinct summary of the main conclusions of our manuscript. While we would agree that there is more to learn about the mechanism of SP140 function, it is not entirely clear to us what the reviewer means when they say that more “direct” evidence is needed for our claim that Sp140 regulates the IFN response during bacterial infection. We feel that the genetic experiments we provide are clear on this point. The reviewer may be thinking that we are proposing a specific mechanism, e.g., that our model is that Sp140 regulates IFN production by binding to the IFN β gene; although that is an appealing possibility, we agree that is not shown in our manuscript, and indeed, we are careful not to make any such claim. Indeed, we explicitly state that a more indirect mechanism is possible (line 390). What is clear, though, is that loss of Sp140 mediates susceptibility to infection via (direct or indirect) increases in type I IFN. We observe increased type I IFN responses in Sp140^–/–^ mice in vivo, and moreover, we find that a cross of Sp140^–/–^ mice to Ifnar^–/–^ mice reverses susceptibility to infection. These results demonstrate that the dysregulation of type 1 IFN in the absence of Sp140 is not merely correlative, but in fact drives susceptibility to bacterial infection in vivo.

Reviewer #3:In this manuscript Ji et al. carefully examine candidate genes driving a previously described susceptibility within the severe susceptibility to tuberculosis (sst1). Surprisingly, mice deficient in the original candidate gene within this locus, SP110, showed no change in susceptibility to infection with M. tuberculosis. In contrast, the authors found that loss of a second gene in this locus, SP140, recapitulated many phenotypes seen in the SST1 mouse, including increased Type I IFN. SP140 susceptibility was reversed by blocking these exacerbated type I IFNs, similar to SST1 mice. RNAseq analysis identify changes in pro-inflammatory cytokines and type I IFNs. The strengths of this paper are the careful and controlled experiments to target and analyze mouse mutants within a notoriously challenging region with homopolymers. Their results are robust, convincing and will be of broad interest to the field of immunology and host-pathogen interactions. Convincingly identifying a single gene within this region that recapitulates many aspects of the SST1 mouse is very important. While a minor weakness is the lack of any mechanistic understanding of how SP140 functions, this is overcome by the impact of the other findings and it is anticipated that this mouse will now be a key resource to dissect the mechanisms of susceptibility in much greater detail.

We thank the reviewer for their generous evaluation. Mechanistically, we do show that Sp140 affects resistance to bacterial infection via regulation of the interferon response, which we think is an important and technically non-trivial advance that provides insight into the function of Sp140. However, we agree that the mechanism for how Sp140 regulates type I IFN is not shown (nor is it claimed to be shown) and addressing this mechanism is now an important and exciting question for future studies.